# Domain wall magnetic tunnel junction-based artificial synapses and neurons for all-spin neuromorphic hardware

Long Liu [1,2], Di Wang [1,2], Dandan Wang [3] ✉, Yan Sun [4], Huai Lin [1,2], Xiliang Gong[4], Yifan Zhang[1,2], Ruifeng Tang [1,2], Zhihong Mai[3], Zhipeng Hou [5], Yumeng Yang [6], Peng Li [7], Lan Wang [8], Qing Luo [1,2], Ling Li[1,2], Guozhong Xing [1,2] ✉ & Ming Liu [1,9] ✉

We report a breakthrough in the hardware implementation of energy-efficient all-spin synapse and neuron devices for highly scalable integrated neuromorphic circuits. Our work demonstrates the successful execution of all-spin synapse and activation function generator using domain wall-magnetic tunnel junctions. By harnessing the synergistic effects of spin-orbit torque and interfacial Dzyaloshinskii-Moriya interaction in selectively etched spin-orbit coupling layers, we achieve a programmable multi-state synaptic device with high reliability. Our first-principles calculations confirm that the reduced atomic distance between $5d$ and $3d$ atoms enhances Dzyaloshinskii-Moriya interaction, leading to stable domain wall pinning. Our experimental results, supported by visualizing energy landscapes and theoretical simulations, validate the proposed mechanism. Furthermore, we demonstrate a spin-neuron with a sigmoidal activation function, enabling high operation frequency up to 20 MHz and low energy consumption of 508 fJ/operation. A neuron circuit design with a compact sigmoidal cell area and low power consumption is also presented, along with corroborated experimental implementation. Our findings highlight the great potential of domain wall-magnetic tunnel junctions in the development of all-spin neuromorphic computing hardware, offering exciting possibilities for energy-efficient and scalable neural network architectures.

As a new paradigm for parallel processing, neuromorphic computing (NC) has attracted intensive attention worldwide due to the great potential in artificial intelligence (AI) and big data analysis applications with overwhelming performance than the conventional von Neumann architecture[1–5]. Deep neural networks (DNNs), mimicking the biological structure and working principles of human brains, have been widely applied in various areas such as image[6,7], speech[8] and video recognition[9], and data classification[10], demonstrating superior

[1]Key Lab of Fabrication Technologies for Integrated Circuits, Institute of Microelectronics, Chinese Academy of Sciences, Beijing 100029, China. [2]University of Chinese Academy of Sciences, Beijing 100049, China. [3]Hubei Jiufengshan Laboratory, Wuhan, Hubei 430206, China. [4]Shenyang National Laboratory for Materials Science, Institute of Metal Research, Chinese Academy of Sciences, Shenyang 110016, China. [5]Institute for Advanced Materials, South China Normal University, Guangzhou 510006, China. [6]School of Information Science and Technology, ShanghaiTech University, Shanghai 201210, China. [7]School of Microelectronics, University of Science and Technology of China, Hefei 230026, China. [8]Lab of Low Dimensional Magnetism and Spintronic Devices, School of Physics, Hefei University of Technology, Hefei 230009 Anhui, China. [9]Frontier Institute of Chip and System, State Key Laboratory of Integrated Chips and Systems, Zhangjiang Fudan International Innovation Center, Fudan University, Shanghai 200433, China. ✉e-mail: wangdandan@jfslab.com.cn; gzxing@ime.ac.cn; liuming@ime.ac.cn

advantages among the applications requiring unprecedently increased speed and capacity for training huge data sets. In general, DNNs consist of multiple layers connected through synapses with updated weights. The summation of products of inputs and corresponding synaptic weights is calculated first and then applied with an activation function to get the output of such specific layer which could further act as the input to the next layer. Synapses with linear weight modulation and neurons with non-linear activation functions are basic elements of most importance constructing a neural network[11]. Explicitly, compact hardware implementation of efficient and reliable bio-inspired synapse and neuron devices is one of the major challenges limiting the development of the NC chip[2,12].

Notably, hardware implementation of neuromorphic devices based on emerging nonvolatile memories (NVMs) offers significant performance advantages when combined with traditional complementary metal-oxide semiconductor (CMOS) technology. The integration of NVMs and CMOS electronics provides benefits such as nonvolatility, scalability, direct mapping of synaptic weights, as well as facilitating functions such as data thresholding, conversion, and trimming required for each layer of a neuromorphic DNN[7,13–18]. There are typical reports on the realization of operations in DNNs using different types of NVMs, such as resistive random-access memory (RRAM)[19,20], phase change memory (PCM)[21], ferroelectric RAM (FeRAM)[22], flash memory[23], and magnetic RAM (MRAM)[24]. While these NVMs show promise for neural network applications, they also come with inherent challenges related to nonlinearity, energy efficiency, area overhead, and reliability[3]. These challenges make it difficult to customize the NVMs, resulting in a loss of learning accuracy and hindrances in implementing specific either synaptic or non-linear activation functions. These issues pose significant confrontations for the practical implementation of NVMs in neural network applications[25]. Nevertheless, each type of NVM possesses its own advantages and disadvantages, with some being slower, having limited endurance, larger area footprints, or only supporting 1-bit operations. There is pressing need to explore the uniqueness among different memory features and synergistic integration with CMOS in order to achieve optimal performance in neuromorphic DNN computing. Importantly, spintronic devices with rich, reproducible and controllable magnetization dynamics, which can emulate functions of synaptic and various types of neurons[25–36], have been receiving increasing attention worldwide in recent years. Among the spintronic devices, the domain wall (DW) magnetic tunnel junction (MTJ) leveraging DW dynamics, which can be precisely manipulated by all electrical methods[37–41] as an information token, is an ideal candidate for application to both linear weight updating and nonlinear activation functions in neural networks because of its intrinsic linear relationship between junction magnetoresistance and programing stimulus. Particularly, in recent demonstrations of CoFeB/MgO based MTJs, the tunnel magnetoresistance (TMR) ratio has significantly improved (>200% at room temperature) while keeping read and write voltage at relatively low values (-0.5 V), manifesting the promise of this technology[42,43].

In the present work, a distinct type of DW perpendicular MTJs (pMTJs) based multi-state synaptic device is experimentally demonstrated, in which a series of DW pinning centers (PCs) is introduced by selective etching of the spin-orbit coupling (SOC) layer to tailor the interfacial Dzyaloshinskii−Moriya interaction (iDMI) strength in the PC regions. The uniformly spaced PCs result in controllable and stable multi-states that can be linearly manipulated by a magnetic field or pulsed electrical current. Next, a novel sigmoid activation function generator is explored based on the same design scheme and fabrication process flow as the developed synaptic device. PCs in the activation function generator are non-linearly distributed which results in a sigmoidal-like resistance state switch driven by the synergetic effect of spin-orbit torque (SOT) and tunable iDMI as elaborated by extensive first-principles ab initio investigations. The systematic experimental and theoretical calculation results with micro-magnetic and circuit-level co-simulations complementarily verified the feasibility of the proposed sigmoidal activation function generator. It is essential to emphasize that the present research primarily focuses on advancing spintronic components, specifically spintronic synapses and neurons, within a larger framework that integrates CMOS electronics. Our DW-pMTJs based all-spin synaptic and sigmoidal neuron prototype shows great potential in energy-efficient neuromorphic hardware development with high performance in a standard CMOS-process-technology-compatible way.

## Results

### Controllable iDMI modulation in heavy-metal/ferromagnet (HM/FM) heterostructure

Figure 1a illustrates the hysteresis loop of W($t$)/Co$_{20}$Fe$_{60}$B$_{20}$(0.9)/MgO(2)/W(3)/Ru(2) (unit: nm) film-stack samples measured by the vibrating sample magnetometry (VSM), suggesting strong perpendicular magnetic anisotropy (PMA) from the grown samples with all different W thickness ($t_w$) ranging from 3 to 5.2 nm. The saturation magnetization ($M_s$) and PMA constant ($K_u$) were derived from the magnetic hysteresis loops data as shown in Fig. 1b, exhibiting a non-linearly upward trend with $t_w$. The film stacks were subsequently patterned into Hall bar structures with dimension of 15 μm × 50 μm as the current channel by dedicated photolithography and ion milling techniques to determine the iDMI strength using current-induced hysteresis loop shift method[44]. An in-plane (IP) $H_x$ along the current direction with a fixed value of −1584 Oe was applied firstly as depicted in Fig. 1c, and then swept the out-of-plane (OOP) magnetic field $H_z$ back and forth between −300 and 300 Oe, while measuring $R_{xy}$ based on anomalous Hall effect (AHE) using an elevated $I$ of ±8 and ±4 mA ($J_e$: 6.6×10$^7$ - 1.3 × 10$^8$ A/cm$^2$). A clear loop shift of $H_z^{eff} = (H_{c1} + H_{c2})/2$ was obtained and illustrated in Fig. 1c, where $H_{c1}$ and $H_{c2}$ are the coercivity fields derived from the hysteresis loops. Figure 1d shows a linear relation between $H_z^{eff}$ and $I$, and the slope refers to the SOT efficiency ($\chi$). Consequently, the systematic $\chi$ dependence on $H_x$ of samples with different $t_w$ is demonstrated in Fig. 1e by sweeping $H_x$ and repeating the same procedure with the critical saturating $H_x$ equal to $H_{DMI}$ in quantity[44]. Figure 1f demonstrates iDMI constant $D$ ($D = \mu_0 H_{DMI} M_S \Delta$) dependence on $t_w$, where $M_s$ is the saturation magnetization of the CoFeB layer (in emu/cc) and $\Delta$ the DW width obtained from $\Delta = \sqrt{A_{ex}/K_{eff}}$, $A_{ex}$ is the exchange stiffness (in pJ/m), $K_{eff}$ is the effective anisotropy energy (in kJ/m$^3$). The value of $D$ exhibits a monotonous decrease as $t_w$ increases within the range of 3.6–5.2 nm. This observation is consistent with findings reported in previous studies[45–50] involving similar W/CFB heterojunction system as shown in the inset of Fig. 1f indicated by the dashed line.

The iDMI dependence on W thickness in W/CFB structure is also verified by our extensive first-principles calculations as demonstrated in Fig. 2. Based on the fundamental calculation cases[51–53] to unveil mechanism of iDMI in HM/CoFeB interface from experiments, Co-terminated configuration in W/CoFe was adopted in our calculations. The left panel of Fig. 2a shows the orientation determination of the iDMI vector from the local geometry in FM/HM heterojunction interface. The specific W/CoFe structure is modeled and depicted in the right panel of Fig. 2a, where a dedicated interface was formed between a layer of CoFe and the surface of body centered cubic (bcc) W (001) with a thickness ranging from 3 to 7 monolayer (ML). The atomic total iDMI strength $d^{tot}$ was derived by a chirality-dependent total energy difference approach as,

$$d^{tot} = (E_{CW} - E_{ACW})/m \tag{1}$$

where $E_{CW}$ and $E_{ACW}$ corresponds to the energy of clockwise and anticlockwise spin configurations, respectively, as shown in Figure S1. And, $m$ depends on the wavelength of the cycloid[54] which is 8 in our

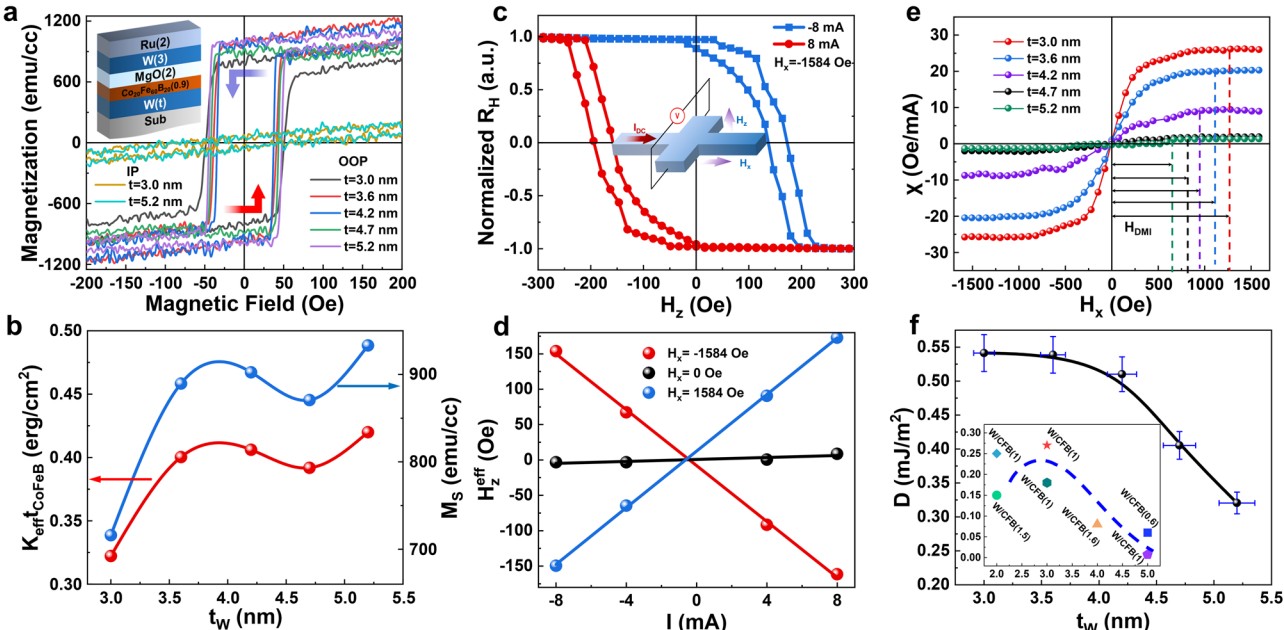

**Fig. 1 | *i*DMI dependence on thickness of heavy metal layer. a** OOP magnetic hysteresis loops of W(*t*)/CFB(0.9)/MgO(2)/W(3)/Ru(2) film stack measured by VSM. The arrows indicate field sweeping direction. **b** $K_u$, $M_s$ dependence on $t_W$ derived from VSM data in **a**. **c** AHE loops with read DC I = ± 8 mA and an in-plane bias field of $H_x$ = −1584 Oe. Inset shows the measurement schematic. **d** $H_z^{eff}$ as function of *I*

under different $H_x$. **e** SOT efficiency χ versus $H_x$, and **f** *i*DMI constant obtained from samples with different $t_W$, the corresponding error bars associated were defined experimentally. Inset shows experimentally measured D in W/CFB heterojunctions adapted from literature reports. The dash line is for guidance of eyes.

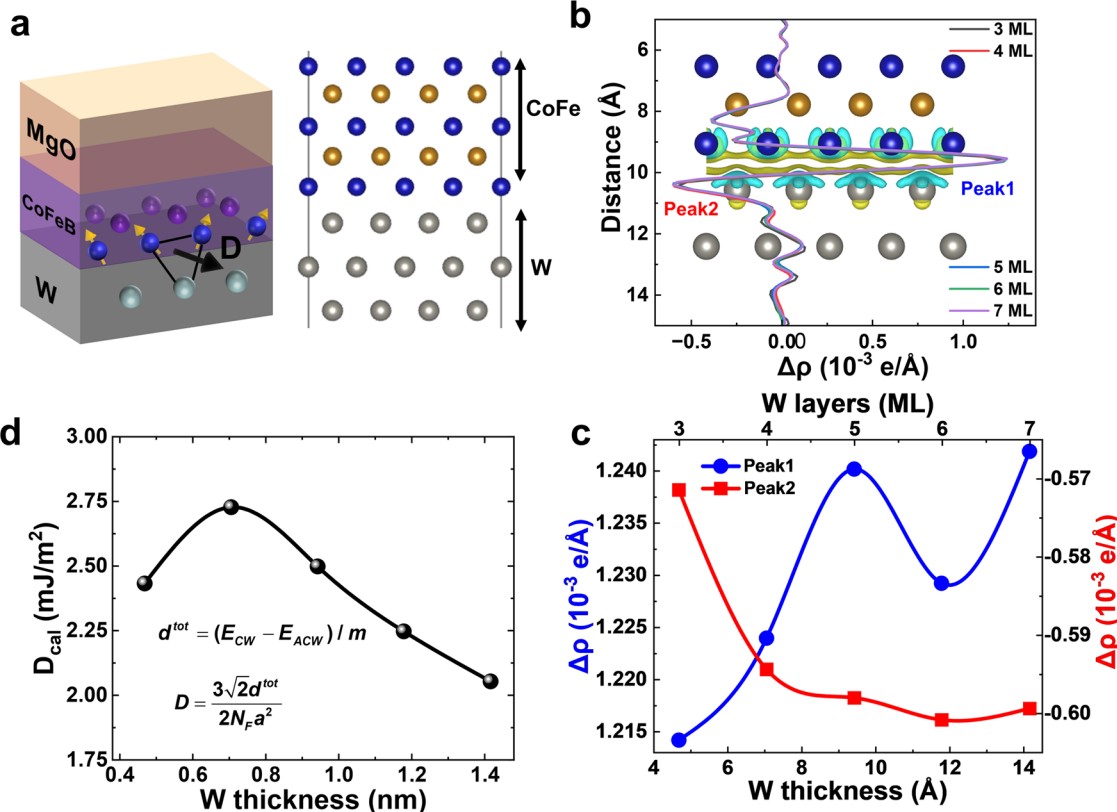

**Fig. 2 | First principles calculations of *i*DMI in W/CoFe heterojunction.
a** Schematic diagram of three sites mechanism of *i*DMI (left panel) and crystal structure of W/CoFe bilayer (right panel). **b** Plane-averaged electron difference

density Δρ(r) (per unit cell) showing the charge transfer in CoFe/W structure with different $t_W$. **c** Gain (peak1) and loss (peak2) of electron peaks extracted from **b** with difference W thickness. **d** Theoretically calculated D dependence on W thickness.

structure as derived in Supporting Information. The atomic total $i$DMI strength of the bilayer can be expressed by the micromagnetic energy per volume unit of the magnetic film[54],

$$E = D\left[m_z \frac{dm_x}{dx} - m_x \frac{dm_z}{dz}\right] \qquad (2)$$

where the constant $D$ can be derived by $d^{tot}$,

$$D = \frac{3\sqrt{2}d^{tot}}{2N_F a^2} \qquad (3)$$

In which $N_F$ is the number of magnetic layer and $a$ is the distance between the nearest neighboring FM atoms (Fe or Co).

$i$DMI strength in FM/HM structure is closely related to interfacial asymmetry which can be modified by surface dipole moment resulted from the charge transfer between HM and FM overlayer[55]. Therefore, in our work the plane-averaged charge density difference $\Delta\rho(r)$ was employed to visualize the charge transfer upon the formation of W/CoFe interface,

$$\Delta\rho(r) = \rho_{tot}(r) - \rho_{CoFeB}(r) - \rho_W(r) \qquad (4)$$

where $\rho_{tot}(r)$, $\rho_{CoFeB}(r)$, $\rho_W(r)$ denote the charge density distributions of the whole system, CoFe and W overlayer, respectively. The results for W/CoFe structure with difference W thickness is shown in Fig. 2b. $\Delta\rho(r)$ is in the vicinity of the W/CoFe interface, a localized charge distribution in the form of a simple dipole is observed, where electrons are mainly gained (lost) at the CoFe (W) side. The analysis reveals characteristic electron gain (peak 1) and loss (peak 2) peaks near the interface, which are integrated and plotted against the thickness of W in Fig. 2c. The magnitudes of these gain and loss electron peaks generally increase with increasing W thickness, indicating a stronger charge transfer and implying a reduction in the strength of the $i$DMI. This trend can be attributed to the increasing surface dipole moment, which enhances interface symmetry. As the charge transfer increases, the resulting larger dipole moment leads to a reduction in $i$DMI strength. Importantly, the larger charge transfer-induced dipole alters the spin-orbit coupling and affects the $i$DMI magnitude. These findings provide valuable insights into the underlying mechanisms governing the $i$DMI behavior in the system under investigation[55,56].

The theoretical analysis reveals a nonmonotonic variation of the $D$ as the thickness of W increases, as depicted in Fig. 2d. Specifically, the results demonstrate an initial enhancement in $D$ when the W thickness increases from 3 monolayers (ML) to 4 ML, followed by a subsequent reduction as the W thickness further increases to 7 ML. This observed correlation is consistent with the behaviors depicted in Fig. 2c and is in qualitative agreement with the experimental findings obtained from our W/CFB/MgO film stacks, as summarized in Fig. 1f. The alignment of $i$DMI with the thickness of W is supported by experimental data, first principles calculations, and existing literature. This substantiates the viability of regulating $i$DMI through the control of W thickness, thereby providing a straightforward yet efficient approach to manipulate DW dynamics[57,58] in all-spin NC hardware for information processing.

**Spin-synapses implementation based on DW-MTJs with tailoring $i$DMI**

Figure 3a shows the specific films structure consisting of substrate/Ta(5)/Pt(1)/[Co(0.3)/Pt(0.3)]$_5$/Co(0.46)/Ru(0.4)/Co(0.6)/W(0.3)/CoFeB(0.8)/MgO(1.1)/CoFeB(1.3)/W(5)/Ru(1.5) in sequence deposited by a magnetron sputtering tool. Numbers inside parentheses refer to the thickness of each layer in nanometers. The film stack adopts a bottom-pinned structure, where the top 5 nm-thick W layer serves as the SOC layer to generate SOT modulating magnetization of the free

layer. The high-angle annular dark-field (HAADF) and high-resolution transmission electron microscopy (HRTEM) images in Fig. 3b indicate that a high-quality smooth and continuous films stack is obtained. The sharp magnetic hysteresis loops within a field range of ±7500 Oe along OOP in Fig. 3c suggest the strong PMA and good quality of free layer (FL), reference layer (RF) and the synthetic antiferromagnet (SAF). The film was then patterned into MTJ pillars with radius of 50 nm by electron beam lithography (EBL) and ion beam etching (IBE) for extensive transport measurements. As validated by repeatable testing cycles, the measured $R$-$H$ loops of patterned MTJ in Fig. 3d corroborate a high TMR ratio up to 145% which is consistent with the values of blanket film measured by the current-in-plane tunneling method. An extensive test was conducted on 100 EBL and UV-lithography patterned MTJs to evaluate their TMR mappings performance. All above extensive characterization results demonstrate the high quality of the grown film stack as well as low edge damage during etching and following process steps (for more information, please refer to Supporting Information Note S1 and Figure S2). The scalability of nano-sized MTJs has been successfully achieved. However, a comparative analysis shows that MTJs with dimensions of 50 μm × 2 μm exhibit superior uniformity at the array level compared with EBL devices as demonstrated in Supplementary Fig. S3. Furthermore, aiming at intuitive multi-states modulation in DW-MTJs with magnetooptical signals visualization, we have selected the micro-sized DW-MTJs as the primary subject of our subsequent investigations.

The prototype SOT-DW-pMTJ-based synaptic device is demonstrated in Fig. 4a with a three-terminal cell structure and schematic diagram of measurement configuration. The write current is applied between two top electrodes (TEs) and generates SOT to drive DW nucleation and motion, while the resistance state is read out by a read current flowing between the TE and bottom electrode (BE), following the equation form[25]:

$$R_{MTJ} = R_P\left(\frac{x_0}{L}\right) + R_{AP}\left(1 - \frac{x_0}{L}\right) \qquad (5)$$

where, $x_0$ is the final position of DW in the stripe MTJ of length L, and $R_P$ ($R_{AP}$) is the resistance of the stripe MTJ when the magnetizations of the free and reference layers are aligned in parallel (antiparallel).

Two ends with size of 50 μm × 50 μm in diamond shape at stripe MTJ device serve as the DW nucleation pads[59], and the channel with 50 μm length and 2 μm width in between works for the DW motion track. The core design of our proposed device lies in the uniquely defined W regions which are partially etched using standard photolithography and dedicated IBE methods, shown as the two slots with 2 μm length and 2 μm width on the heavy metal (HM) in the upper panel of Fig. 4a. According to the relationship of $i$DMI and $t_W$ unveiled in Fig. 1f, the dedicatedly etched regions hold a larger $i$DMI with thinner W thickness due to the ion milling introduced local compression stress at the W/CoFeB interface[60,61], which reduces distance between $5d$ and $3d$ atoms at the interface leading to an enhanced $3d$$-5d$ orbital hybridization[62–64]. It is noteworthy that the DW energy density ($\sigma_{DW}$) is closely related to $i$DMI strength[65],

$$\sigma_{DW} = 4\sqrt{A_{ex}K_{eff}} + 2K_D\Delta - \pi|D| \qquad (6)$$

The last three terms on the right-hand side of the equation are Bloch wall energy density, DW anisotropy energy density, and $i$DMI contribution, respectively. $A_{ex}$ is the exchange stiffness, and $K_{eff} = K_u - \mu_0 M_s^2/2$ is the magnetic anisotropy. $K_D = N_x\mu_0 M_S^2/2$ and $\Delta = \sqrt{A_{ex}/K_{eff}}$ are DW shape anisotropy and DW width, respectively. From Eq. 6, it can be concluded that the larger $i$DMI in the etched site with thinner W corresponds to lower DW energy when DW stays at that region, equivalent to an energy potential well with a depth of $\pi|\Delta D|$[58], where $\Delta D$ is the difference between the two regions (see below). Therefore,

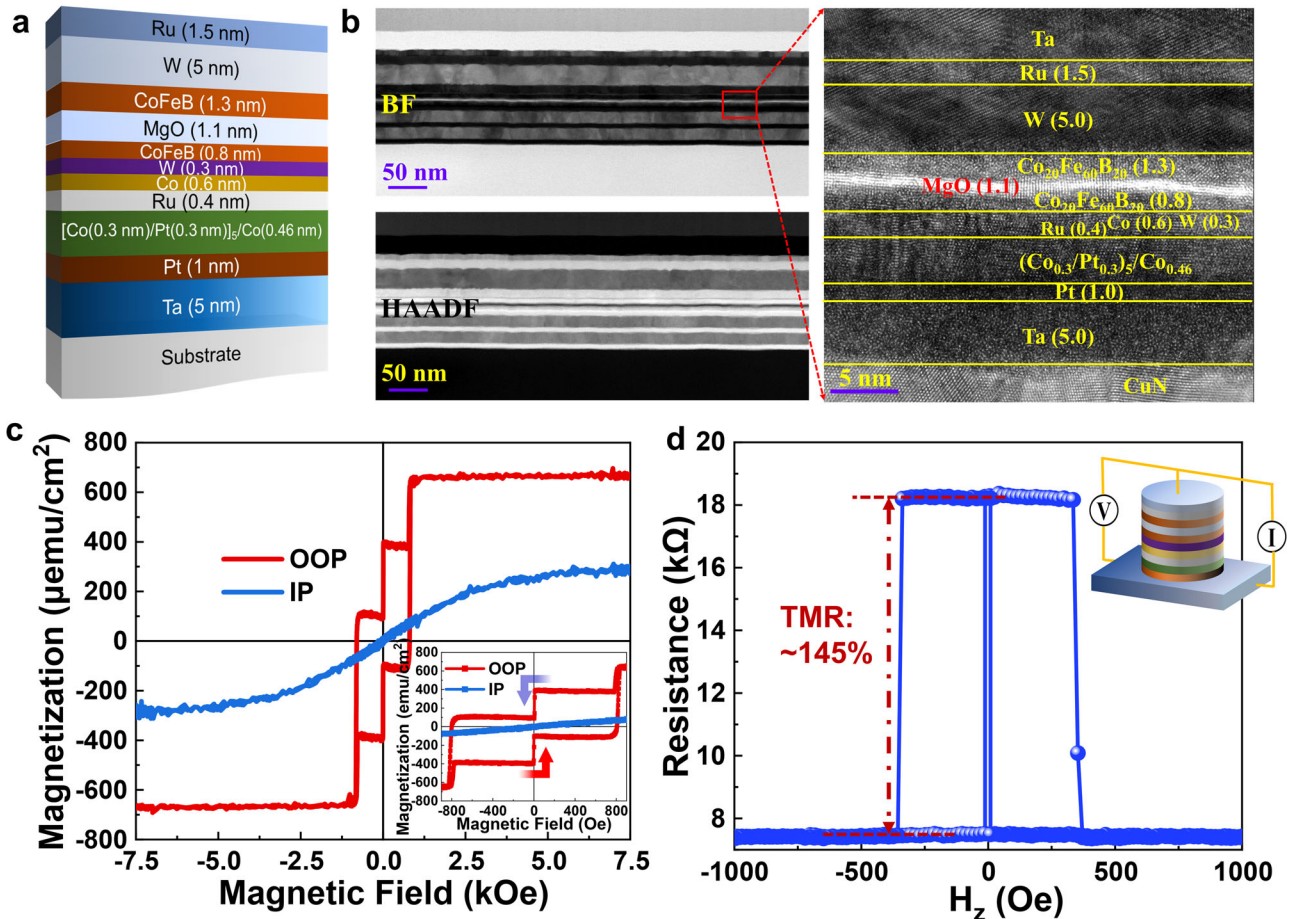

**Fig. 3 | p-MTJ film stacks characterization, devices fabrication and testing. a** Schematic of PMA film stack, and **b** corresponding HRTEM bright-field (BF) and HAADF images. **c** OOP and IP *M-H* loops measured by VSM. Inset shows magnified loops at low fields. **d** Magnetoresistance versus $H_z$ of p-MTJ device.

the partially etched W regions with tailored *i*DMI can serve as reliable DW PCs, enabling accurate and effective DW motion manipulation.

Clearly, for the single device and devices array with two engineered PCs formation as shown in Fig. 4a, four discrete resistance states (I - IV) of DW-pMTJ were achieved and switched by SOT-driven DW motion with an assistive $H_z$ magnetic field. To extensively characterize the devices stack structure and quality, we conducted high-resolution/scanning transmission electron microscopy (HRTEM/STEM) and energy dispersive spectrometer (EDS) measurements and studies. Importantly, the etched profile upon dedicated IBE process between PC edge transition and bottom regions are contrastively formed as illustrated in upper panel TEM images of Fig. 4b; the corresponding HRTEM image in PC bottom region with inverse fast Fourier transforms (FFT) patterns corroborate the alpha phase of W formation in the interfaces. The complementary bright-field STEM and EDS analysis revealed a flat interface with atomic-level high quality of expected films stack structure in Figures S4, S5, and Note S2. Figures 4c and 4d refer to *R-H* loops of the MTJ device before and after PCs etch process, respectively. The measured low TMR ~ 14% (much small than that of pristine DW-pMTJ device ~145%) is attributed to the larger device size and the two-probe magneto-resistance measurement, which can be improved by device scaling and measurement optimization in the future[66]. Transport measurements were carried out upon the application of a large saturating magnetic field (3 kOe) along the -*z* direction. In Fig. 4c, the transport measurement data match well with the Kerr signals, and the sharp flips of *R-H* loop were observed at −756 Oe (726 Oe) and 235 Oe (−235 Oe) for $H_z$ sweeping from −1.5 kOe (1.5 kOe) to 1.5 kOe (−1.5 kOe) which refer to reference and free layer flips, respectively, as indicated by specific-colored arrows. The sharp

flips suggest that there is no evident DW pinning effect in the pristine DW-pMTJ. In contrast, the step-like spin flip behaviors of the free layer were observed clearly in the DW-pMTJ after the PCs etch process, as marked by a green dashed box in Fig. 4d. Figure 4e shows the magnified *R-H* loop marked by green dashed box in Fig. 4d, and the dashed arrows indicate the $H_z$ scan direction. Four evident steps in Fig. 4e corresponding to four stable resistance states corroborate that the effective DW PCs were successfully formed by using the developed selectively partial-W-etch process. A critical magnetization switching field ~6.5 Oe of the free layer in DW-pMTJ with PCs reflected in Fig. 4e is much smaller than that of unetched case in Fig. 4c. This is attributed to defects introduced by ion milling which reduced the DW nucleation energy barrier[67]. Notably, the controllable multi-states modulation driven by the magnetic field was also accomplished as exemplified in Fig. 4f and animated in Supplementary Movie S1. Briefly, we first set the MTJ to state "I" as the initial state after applying a positive saturating magnetic field (3 kOe) followed by a small negative field (−200 Oe). Next, the field was swept from −200 Oe to 300 Oe and the resistance switched from State "I" to State "II". Finally, the field was swept back to −200 Oe, and the resistance switched back to the initial state "I" subsequently. Then we repeated the above operations, but the field was swept to 350 Oe and then swept back, so we obtained the loop of the State "I-III-I" switching trajectory as marked in green in Fig. 4f. The operation of "I-IV-I" switching was in the same manner and is marked in blue in Fig. 4f. The state-by-state switching operation demonstrated in Fig. 4f indicates that the multi-states achieved by the selectively partial-W-etch process are highly stable, controllable, and repeatable, which is also verified by our extensive DW pinning stability studies with a 10-month time frame since fabrication (Supporting Information

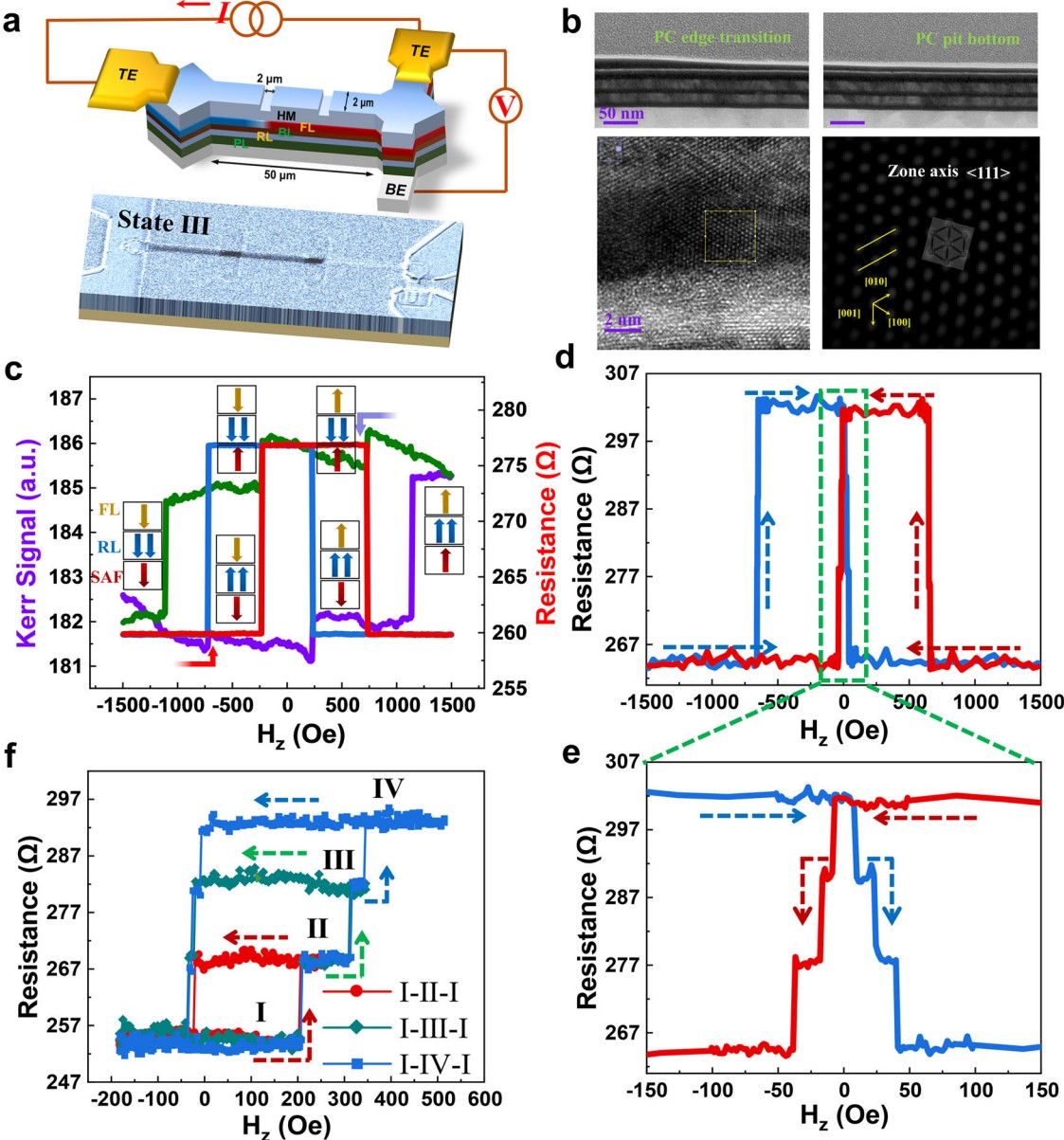

**Fig. 4 | Spin-synapse devices based-on DW-pMTJs. a** Schematic diagram of measurement configuration of four-state synaptic device, representative MOKE image of State III enclosure with magnified PCs. **b** TEM images of etched profiles at PC edge transition and bottom regions and the corresponding HRTEM image in PC bottom with inverse FFT patterns. **c** *K-H* (purple-green) and *R-H* (blue-red) loops as a function of $H_z$ swept (forth-back) from pristine device before PCs formation. Blue curve denotes Kerr signal measured by p-MOKE. Colorful arrows represent respective magnetization orientation of PL, RF, FL at different states. **d** *R-H* loop as a function of $H_z$ of the synaptic device with PCs. Dashed arrows illustrate sweep direction of $H_z$. **e** Magnified *R-H* loop as marked by green dash-box in **d**. **f** State-by-state switching of developed four-state synaptic device. Red, green and blue curves signify state1 (I) to state2 (II), state1 (I) to state3 (III), state1 (I) to state4 (IV) switching, respectively.

Figure S6). The linear multi-states of the proposed DW-pMTJ are direct mapping and motivate further studies for reliable emulation of the linear operation of a biological synapse.

### Spin-neuron realization in the form of sigmoidal activation function generators

As demonstrated in Fig. 5a and Supporting Information Figure S7, the proposed and experimentally executed sigmoidal activation function generator shares the same structure and fabrication process flow as that of the synaptic device. The core difference between these two types of devices is the dedicately configured gaps distribution of PCs. For synaptic devices, PCs are uniformly distributed to achieve linear multi-states for linear weights manipulation. In contrast, for the sigmoidal activation function generator, the PCs are non-linear distributed based on piecewise approximation. Since the change of resistance for neighboring states is proportional to the space length of corresponding PCs, the designed sigmoidal activation function generator resistance variation can be a sigmoidal function of the states switching driven by the SOT current or $H_z$ pulse.

Driven by a dedicately programmed pulse $H_z$ field applied by electromagnet utilizing a current ranging from tens to a few hundreds of milliamperes, the remaining energy after DW overcomes the energy barrier is varied for different state switching, leading to a diverse traveling distance of the DW after turning off the $H_z$ pulse. It was experimentally corroborated by the different magneto-resistance states with corresponding DW position Kerr signatures as depicted in Fig. 5a, b under the measuring protocol of sigmoid

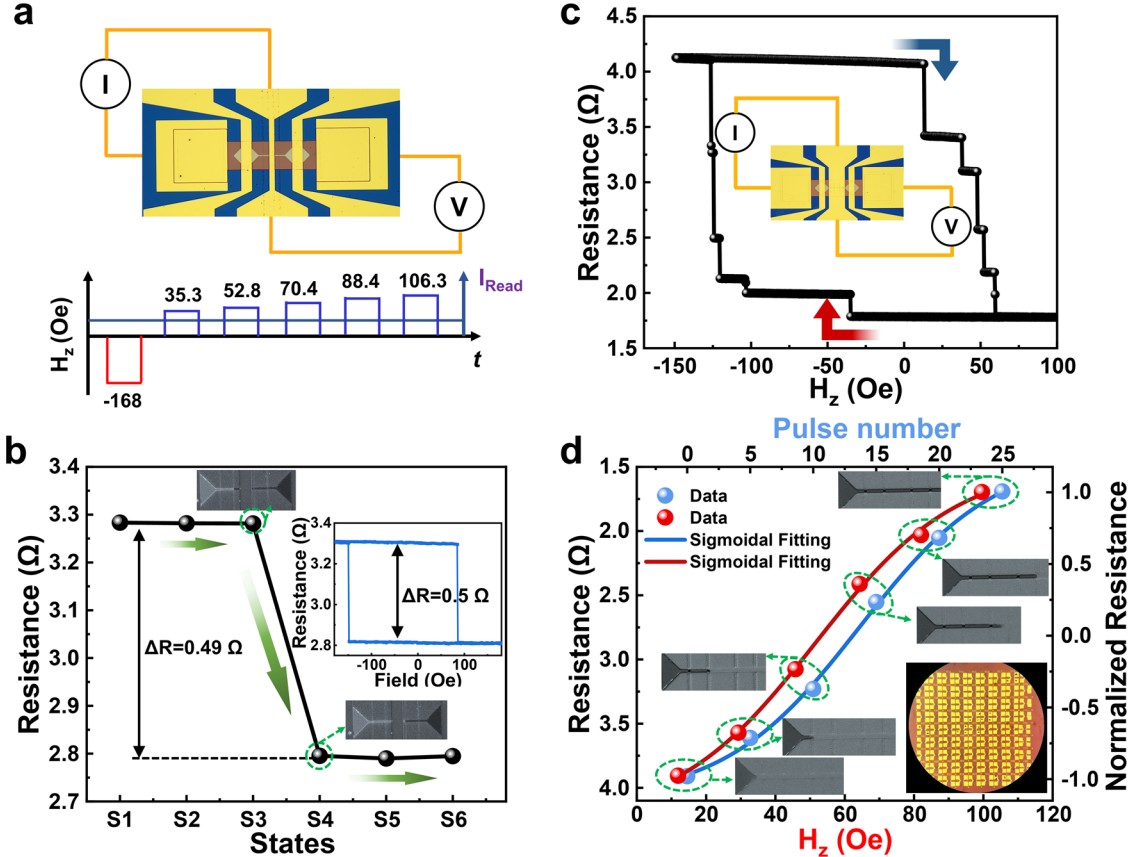

**Fig. 5 | Spin-neuron based on DW-pMTJ sigmoidal activation function generator. a** MOKE image, the measuring configuration and protocol of spintronic sigmoid neuron based on DW-pMTJ. **b** Different resistance states with corresponding DW position. The green arrows indicate the state switching direction. Inset illustrates the resistance curve *vs* sweeping $H_z$. **c** Resistance as a function of $H_z$ of sigmoidal device with central top electrode covering the whole channel (inset image). The arrows indicate field sweeping direction. **d** Resistance as a function of the amplitude and numbers of pulsed magnetic field for sigmoidal device in **c**. Insets depict the correspond array and magnified photos of the DW-pMTJ sigmoidal devices.

spin-neurons based on DW-pMTJ. The measurement setup and test sequence are illustrated Fig. 5a. Briefly, a four-probe method with a current path between left BE and one port of the partially covering central TE and a voltage path between right BE and another port of the central TE was adopted for transport measurements. A negative $H_z$ pulse of −168 Oe was applied to saturate FL magnetization in down direction, followed by a positive $H_z$ pulse sequence with linearly increasing amplitude to nucleate and drive the up domain. The junction resistance was read out timely right after every single $H_z$ pulse application by a 300 μA read DC. The measured evolution of junction resistance over state switching is shown in Fig. 5b with corresponding MOKE image, which suggests that the DW is driven in an order of PC-by-PC, leading to state-by-state magnetization switching under the positive $H_z$ sequence. Note that only the switching from state 3 to state 4 results in abrupt resistance change, corresponding to the magnetization signal of the segment of FL under the central TE. Thus, it is concluded that the employed measurement setup just precisely detected the magnetization signal of the central TE covering regime, leading to a partial TMR signal ~17% as illustrated the inset of Fig. 5b.

Different from the partially covering central TE device in Fig. 5a, b, the *R-H* loop of device with central TE covering whole channel demonstrates six distinct states and TMR up to 130% as shown in Fig. 5c, which is much higher than TMR result of the synaptic device with two-probe measurement method and close to that of pillar MTJ with radius of 50 nm. Under the similar test sequence to Fig. 5b, one can get the relation of junction resistance versus $H_z$ amplitude as demonstrated in Fig. 5d. The solid black sphere symbols in Fig. 5d

illustrate the dynamic evolution of DW position which is proportional to junction resistance. The results are well fitted with a shifted sigmoid function as illustrated by the red line. The fitting results demonstrate that the profile of DW position with corresponding MTJ magneto-resistance analogs a sigmoidal function of pulsed B-field amplitude. It can also be a function of B-field pulses number or duration analytically. After DW moves into the required PC, a rest pulsed-field with the opposite direction turns on to force the DW back to its initial position for resistance state initialization. Note that the switching order might require to be re-organized according to the designed sequence to realize the sigmoid function. However, it is noted that such adjustments can be made without compromising the feasibility of the proposed mechanism. Importantly, the MOKE signal becomes elusive as the central TE covering the whole channel, which can be further optimized by introducing Indium Tin Oxide (ITO) TE as shown in the insets of Fig. 5d. To summarize, the complementary experimental results of resistance as a function of $H_z$ of the sigmoidal device with central TE covering partial (Fig. 5b) and whole channel (Figs. 5c, d) elaborates the correlation of the state with implemented sigmoidal activation function generator functionality. This is also supported by Figure S7 and Supplementary Movies S1–S6, which demonstrate extensive micromagnetic simulations and MOKE imaging. These investigations explore synaptic and sigmoid neuron signals under various pulsed magnetic field strengths and $J_e$ with different amplitudes and numbers. Importantly, such reliable and controllable DW motion manipulation can be realized by pulsed current via SOT and/or STT effect as reported previously[57,68].

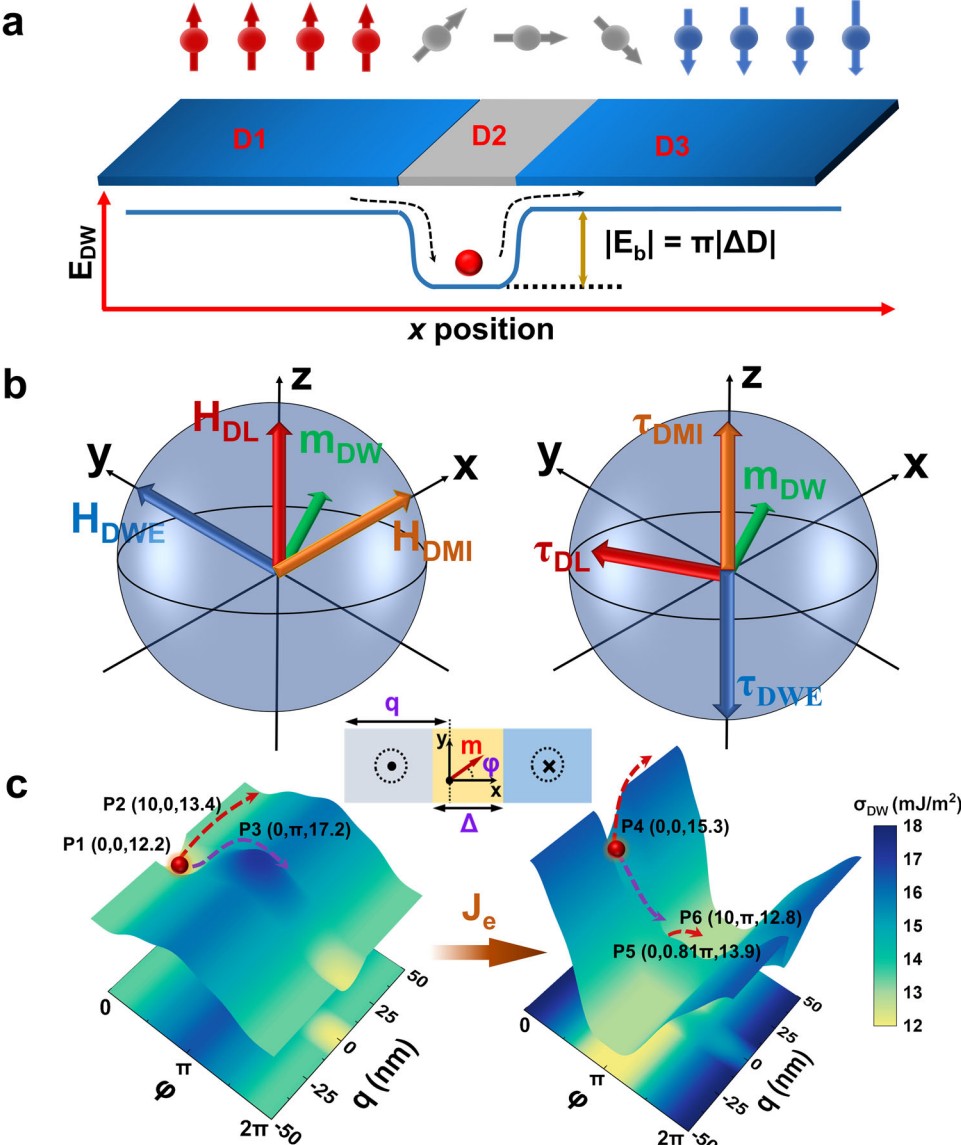

**Fig. 6 | SOT and *i*DMI synergistic effect on DW dynamics with quantitative visualization. a** Schematic of proposed DW pinning mechanism with DMI profile. **b** Effective fields (left) and torques (right) in FM where m$_{DW}$ (green), DL (red), DMI (yellow), DWE (blue) denote center magnetization of domain wall, damping-like term of SOT effect, domain wall energy and *i*DMI, respectively. **c** Quantitative determination and illustration of DW motion and energy landscape with and without $J_e$. Inset depicts the schematic of DW coordinates of $q$ and $\varphi$.

## Qualitative analysis of DW dynamics under SOT and *i*DMI synergistic effect

As schematically elaborated in Fig. 6a of DW pinning mechanism with *i*DMI steps, where $D_2 > D_1 = D_3$, DW energy configuration in prototype device with current injected along $+x$ direction. Figure 6b depicts the field and torque configuration acted on center magnetization of DW (m$_{DW}$). In the present system, the DW-energy-induced torque competes with *i*DMI contribution and tend to convert the DW into Bloch type. The electron current flowing along the $x$ direction induces an anti-damping-like SOT torque rotating m$_{DW}$ towards the $y$ direction, leading the m$_{DW}$ switching into the $z$ direction driven by the torque from the *i*DMI effective field $H_{DMI}$[37,69].

Figure 6c illustrates the visualized energy landscapes with quantitative determination of a DW, with and without $J_e$, respectively, as functions of two collective coordinates (DW position $q$ and angle $\varphi$ tilted from $x$ axis as demonstrated in the inset) based on the 1D DW ($q$, $\varphi$) motion model using the Thiele's approach[70], from LLG equation in the rigid DW limit (i.e., DW width $\Delta = \sqrt{A_{ex}/K_{eff}}$ does not change).

With the force equation[71], as $\frac{\alpha}{\Delta}\dot{q} + \dot{\varphi} = -\gamma\frac{\pi}{2}H_{DL}\cos\varphi$, and torque equation as $\frac{1}{\Delta}\dot{q} - \alpha\dot{\varphi} = -\gamma\frac{\pi}{2}\frac{D}{\Delta M_s}\sin\varphi - \gamma\frac{K_D}{M_s}\sin 2\varphi + \gamma\frac{\pi}{2}H_{FL}\cos\varphi$. The domain wall energy landscape is calculated as[72] $\sigma_{DW} = 4\sqrt{A_{ex}K_{eff} + 2K_D\Delta\cos^2 2\varphi} - \pi D\cos\varphi - \pi\mu_0 M_s\Delta H_{FL}\sin\varphi$. A parabolic type pinning center is formed in the range of $q\in[-10, 10]$ nm with a maximum of $D = 0.8$ mJ/m$^2$, while $D = 0.4$ mJ/m$^2$ outside the pinning center. Other magnetic parameters are consistent with the micromagnetic simulations as listed in Table S1. Without $J_e$ application, there are two minimum energy position, (0, 0) and (0, 2π), marked as P1. A DW moving in the $q$ direction (P1 to P2 indicated by the red arrow dashed) or $\varphi$ direction (P1 to P3 indicated by the purple dashed arrow) must overcome a large energy barrier which suggests that DW is stabilized in the pinning center with a ↑→↓ chirality naturally determined by the unique DMI configuration. However, in the presence of a $J_e$ pulse ($2\times10^8$ A/cm$^2$ in this case), the energy landscape is tilted in the $\varphi$ direction. The DW moves downstream along the energy gradient in the $\varphi$ direction from P1 to P5 (indicated by the purple dashed arrow), referring to DW center magnetization rotation upon the effect of SOT.

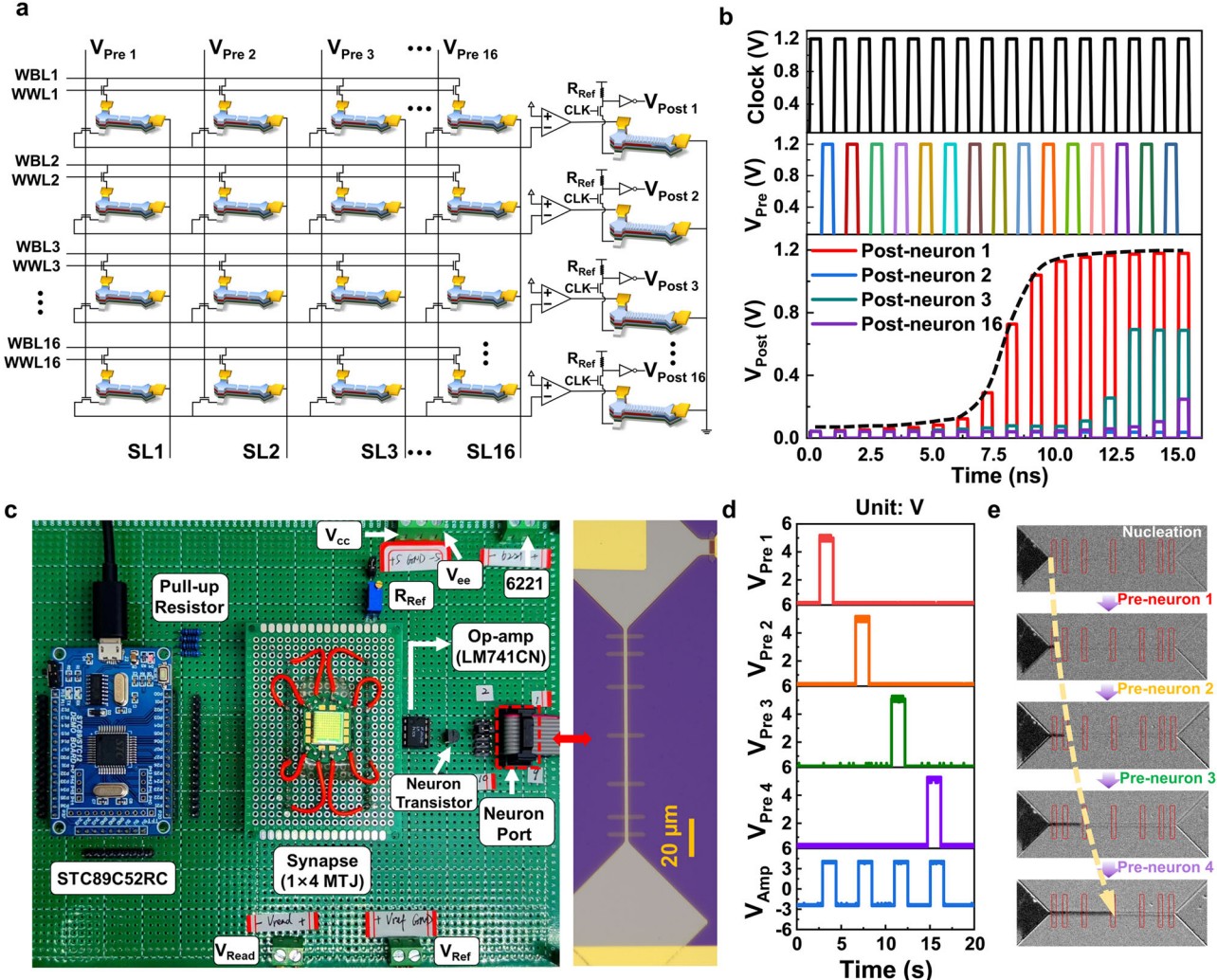

**Fig. 7 | Circuit simulation and experimental verification of spin neuron circuit.**
**a** Schematic of a simplified ANN network comprising 16×16 DW-MTJ synapses and 16×1 16-state DW-MTJ neurons. **b**. Transient simulation results of the spin neuron circuit. **c** Hardware implementation of with developed 4 binary SOT-MTJs as synapse and 1 spin-neuron device (right panel). **d** Waveform of the pre-neuron signal output by MCU and corresponding output of the operational amplifier. **e** MOKE Kerr images captured after each pre-neuron signal applied.

Then the DW gets over the pinning center and relaxes to P6 (the minimum energy position of the whole system). Explicitly, the process is indicated by the red dashed error. Such DW depinning process coincides with the mechanism of DW motion under a synergistic effect of SOT and DMI as unveiled in Fig. 6b.

**Simulation and experimental verification of spin neuron circuit**
A simulation of a hybrid MTJ/CMOS system that incorporated 16 × 16 DW-MTJ synapses and 16 × 1 16-state DW-MTJ neurons was performed (parameters listed in Table S2). Furthermore, to ensure the reliability of developed devices, we conducted a validation of the system's performance using a neuromorphic hardware implementation of representative arrays. The results of both the simulation and experiments are presented in Fig. 7. The purpose of such complementary study is to assess the feasibility of employing all-spin synapses and a sigmoidal activation function generator, as well as to evaluate the compatibility of our developed devices with CMOS technology. Figure 7a illustrates the simplified artificial neural network (ANN) used in the simulation. Each synapse cell consists of a DW-MTJ-based synapse and two control transistors for write and read control. The gates of these transistors are connected to the word line (WWL) and the pre-neuron signal ($V_{Pre}$), respectively. The neuron cell comprises a DW-MTJ-based activation function generator, which forms a voltage divider with a reference resistor. It also includes a read control transistor and an inverter for output generation. The DW-MTJ-based devices are modeled using Verilog-A, and Supplementary Note S3 provides details on the co-simulation with the CMOS peripheral circuit. For simplicity, the DW-MTJ-based synapses operate in a binary mode with a default weight value.

The transient simulation results presented in Fig. 7b provide insights into circuit-level neuromorphic behaviors. Pre-neuron signal is applied asynchronously to the corresponding column to open all read control transistors in this column. The read current from source line (SL) flowing through DW-MTJ is sent to the operational amplifier generating a weighted current pulse injected into the write channel of the activation function generator to drive the DW motion and state refreshing. Next, the resistance state is read out by the voltage divider controlled by clock signal and amplified by the inverter to get the neuron output, labeled as $V_{Post}$. Since all the synapses connected to activation function generator 1 are in low resistance state, the DW in this generator is driven to next PC in each cycles. The output voltage of post-neuron1 is a non-linear function of pulse number by time in a sigmoidal profile, as indicated by the black dashed guideline.

The experimental construction and characterization of the circuit were carried out, specifically focusing on a representative sub-circuit that contains the all-spin synapse and neuron devices along with the

CMOS part as illustrated in Figs. 7c–e. Such characterization provides a more comprehensive understanding of the device performance and its integration with the CMOS circuitry. The hardware setup (Fig. 7c) consists of one partial array with four binary SOT-MTJs serving as synapses and one proposed neuron device. The synapses are wire-bonded to a PCB carrier, which is connected to a universal board containing an MCU and peripheral circuits. The pre-neuron signals generated by the MCU are measured along with the output of the operational amplifier using an oscilloscope (Fig. 7d).

The experiment includes a sub-circuit of the array with CMOS components, resistors, and the actual device, allowing for the creation of a sample post-neuron. Supplementary Movie 7 provides further detailed correlation. The resulting Kerr images (Fig. 7e) show the progression of the spin-neuron device. A down domain is initially nucleated at the left pad and propelled into the racetrack by current pulses triggered by the pre-neuron signals. The down-up DW is pinned at the corresponding pinning center after each cycle of the pre-neuron signal, causing a switch in the resistance state. This hardware test effectively demonstrates the functional characteristics, integration features, and implementation effects of the developed DW-MTJ all-spin synapses and neurons.

## Discussion

Spurring growth of generative AI models with ever-more-sophisticated devices, hardware implementation of synaptic and neuron devices poises one of the major challenges and constrains the scalability and high-dense integration of neural networks due to the large on-chip area and high-power consumption. We experimentally demonstrated a unique spin synaptic device by introducing a series of effective DW PCs achieved by selective heavy metal partial etching with modulated local $i$DMI. A pioneering proof of concept of dynamic tailoring DW pinning via ion-beam-etching-strain modulated antisymmetric exchange strength is validated at the HM/FM interface in the MTJ. The extensive cross-sectional high-resolution transmission electron microscopy images and energy source of local $i$DMI from first-principles calculations are being conducted to further comprehend the mechanism responsible for $i$DMI engineering. Reliable state-by-state switching corroborates the high controllability, stability, and repeatability of the multi-states introduced by DW pinning and depinning dynamics in the designed PCs. The linear multi-states against the magnetic field are direct mapping of the synaptic function. We further verified the feasibility of constructing a sigmoidal spin-neuron using the same scheme as that of a synaptic device experimentally. By further complementary micro-magnetic and circuit-level co-simulation, a sigmoid activation function generator has been successfully demonstrated based on a DW-pMTJ driven by synergistic SOT and $i$DMI with an energy consumption of 36.3 fJ/pulse. A neuron circuit design with a compact sigmoidal cell area and low power consumption is also presented, along with corroborated experimental implementation. Compared to state-of-the-art counterparts of neural activation function generator, our developed architecture is competitive as shown in Figure S9 benchmark diagram. The overall energy consumption is less than 508 fJ/operation including the reset process with a competitive firing rate up to 20 MHz. Such developed devices are compatible with the current standard CMOS technology and the magnetoresistive random-access memory process, without any exotic material, complicated structure, and extra masks in comparison with the state-of-the-art, offering a promising and applicable candidate for neuromorphic devices and chips application in new trajectories with a device-circuit perspective and the envision design of all-spin neuron circuits. By offering this comprehensive elucidation, it is expected to provide more nuanced insights into the intricate integration of spintronic and CMOS technologies within the realm of neuromorphic computing.

## Methods

### Film deposition and characterization

A series of film stacks were deposited on thermally oxidized silicon Si/SiO$_2$ substrates using DC and RF magnetron sputtering under a base pressure lower than $1.0 \times 10^{-7}$ Torr at room temperature. An annealing process at 350 °C was then applied inside a high vacuum chamber of $4 \times 10^{-8}$ Torr for 1 hour to improve interfacial PMA. Magnetic parameters such as PMA and $M_s$ of the film were evaluated using a vibrating sample magnetometer (VSM) system. The high-resolution transmission electron microscopy (HRTEM) and high-angle annular dark-field scanning transmission electron microscopy (HAADF-STEM) were used to characterize the structure of the cross-section.

### Device fabrication and measurement

The films were fabricated into MTJ devices with a radius of 50 nm by electron beam lithography and ion milling, followed by electron beam evaporated Ti(20 nm)/Au(80 nm) electrodes. All transport measurements were performed in our MagVision MOKE system with a probe station. The current was sourced by Multiple Keithley 6221 and MTJ voltage drop was measured using Keithley 2182 A nanovoltmeter. Both meters were controlled by the MagVision system through scripts.

### Circuit simulation and experimental validation

For the circuit level simulation, a Verilog-A model of a SOT-DW-pMTJ device was developed to co-simulate with the CMOS peripheral circuit using SPICE simulators. A foundry's 28 nm Product Development Kit was used to verify the proposed design. As to the experiment system for spin-neuron sub-circuit implementation, a Labview program in host computer was used to control timing sequence with synchronizing all-spin NC hardware with integration of CMOS components (for more information refer to SI).

## Data availability

The data that support the findings of this study are available from the corresponding authors on request. Source data are provided with this paper.

## Code availability

The authors declare that no code was utilized in the implementation or analysis of the research presented in this manuscript.

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

## Acknowledgements

G.Z.X. acknowledges the funding support of National Key Research and Development Program of China (No. 2021YFB3601300), the National Natural Science Foundation of China (No. 62074164 and 62374180), and the Strategic Priority Research Program of the Chinese Academy of Sciences under Grant No. XDB44010000. L.W. acknowledges the funding support of the National Natural Science Foundation of China under Grant No. 92365113. D.D.W. and P.L. thanks the Hubei Jiufengshan Laboratory, the State Key Laboratory of Integrated Chips and Systems of Fudan University and the Information Science Laboratory Center of USTC for support, respectively. Authors would like to acknowledge the thin films stack samples depositions support from Qingdao Research Institute, Beihang University with fruitful discussions.

## Author contributions

G.Z.X. conceived the core idea of this study. L.L., D.D.W., G.Z.X., H.L., D.W. and Z.H.M. performed the experiment and measurement. L.L., G.Z.X, D.W and D.D.W. performed the micro- and circuit-level simulations. G.Z.X., L.L., D.W, H.L., Y.F.Z., R.F.T., Y.S, X.L.G, Z.W.W, D.D.W., Z.P.H., Y.M.Y., P.L., L.W., Q.L., L.L. and M.L. conducted analysis. L.L., D.D.W. and G.Z.X. prepared the manuscript. All authors contributed to the interpretation of the results and discussions.

## Competing interests

The authors declare no competing interests.

## Ethics Statement

This statement emphasizes our adherence to principles such as diversity, equality, and responsible conduct during the study design, data collection, analysis, and interpretation phases.
