## [Peer Review File · Nature Communications]

REVIEWER COMMENTS

Reviewer #1 (Remarks to the Author):

This paper reports an interesting endeavor to demonstrate a reliable, multi-state synaptic device based on domain wall MTJ and a sigmoid activation function generator. The concept, apart from the function generator scheme, is not new, but this work can be considered one of the first experimental reports in which the programmability and the distinction of states are adequately emphasized. While the experimental part on the device is interesting, the characterization of the circuit, or even a smaller sub-circuit containing the all spin device, along with CMOS part, is lacking and in my opinion that can be the differentiating factor that elevates research of this magnitude to the level of novelty that warrants publication in a prestigious journal like nature communication (more about this in my detailed comments).

Here are my most important concerns about this paper:

- 1- The circuit part of this paper is lacking in my opinion and needs a full revisit. First of all, since this is just a simulation, why did the authors limit the array to 4x4? By using a larger array and a longer pre-neuron pulse train (they can be closer to each other... why are the pulses have such a long separation?) they can showcase a larger range of states.
- 2- What is the purpose of Clock signal shown in fig 7? where is it applied? There's no indication anywhere
- 3- I suppose the Vpost-neuron signal is the node after the inverter, in fig 7a it's not properly placed, implying that it's before the inverter.
- 4- A major problem with the circuit in 7a: If post neuron signal is captured after the inverter, as stated clearly in the manuscript, the inverter is supposed to give "full-swing" output, why do we see non full-swing signals in the last panel of fig 7b? (post neuron 3 and 4, as egamples) This is pretty confusing.
- 5- The authors say "The output voltage with an evident "S" shape indicates that the developed sigmoidal multi-states are successfully transformed into circuit timing. " What do you mean by "s" shape output? I think there's some mistranslation happening here.
- 6- The writing of "spin-neuron circuit implementation ..." section must be improved, it becomes unreadable at some points.
- 7- The calculation (simulation?) results related to power/energy consumption are not clearly shown in the paper and any discussion on energy is limited to Fig s7 in the supplementary. the reported 3.5fj energy is per pulse, but an "operation" does not consist of a single pulse, but multiple pulses. So it must be clarified what the authors mean by "operation" when they report 67 fj/operation. I suggest they add a section in supplementary material explaining how they came up with this number. It's important to consider the difference in the meaning of "operation" in this case in which the device must be "settled" to its desired state, versus a simple logic like an inverter which flips between two definite states. For example, if as mentioned in the paper, it takes 7 cycles for DW to reach the last PC, we are dealing with a

very different type of "operation" and the actual energy consumption can only be calculated by averaging for all combinations of input/outputs and settling times.

8- I strongly suggest that the authors setup a partial experiment and include part of the array, it can be just top row with 4 CMOS, 4 low/high resistors and the actual device, creating a sample post neuron. For me that experiment qualifies this paper for publication.

Reviewer #2 (Remarks to the Author):

The major experimental contributions of this paper lie in acquiring both multi-state memory behaviors ("synapse") and activation function("neuron") by utilizing a spintronic device, in particular, by controlling the distribution of domain-wall pinning centers. I find the experiment and device concept themselves quite exciting. However, this excellent experimental advance is masked and compromised by many exaggerated, non-factual, or illogical statements such as "all-spintronic neuromorphic computing hardware" that appear throughout the manuscript, which are not consistent with, and show the misunderstanding of, how deep neural network (DNN) computing is actually done at the system level. For acceptance, the present form of the manuscript needs a substantial revision so that its logic and contents can be much more robustly structured just on the essence of the work with scientific rigor and scholarly balance.

1. An example of the illogical, non-factual, or exaggerated statement is the first sentence of the second paragraph of Introduction, which is contradictory to how the DNN computing is actually done. That is, NVMs and CMOS electronics are not what should be separated and compared against each other, but what should be used together for neuromorphic DNN computing. Concretely, a DNN consists of many layers, and each layer is a concatenation of the multiply-accumulate (MAC) operations for matrix multiplications, the activation function for decision making or thresholding, and the data conditioning and trimming such as pooling, max-pooling, soft-maxing, and many more. Thus the actual neuromorphic DNN implementation requires, for each layer, not just NVMs to store synaptic weights to perform the MAC operations, but also CMOS electronics to perform data thresholding (ok, in authors case, this may one day be possibly done with the same spintronic device in the "neuron" function), data conversion (necessary due to the analog computation with NVMs), data conditioning, and data trimming. So NVMs and CMOS electronics (mixed-signal and digital electronics) are not two competing technologies, but complementary technologies that should work together for neuromorphic DNN computing (and the industry already embeds NVMs such as RRAMs and MRAMs in commercial CMOS technology). For directly relevant examples as to how CMOS and NVMs must work together or how CMOS and various NVMs are available together through industry, please see

- Nature 577, 641–646 (2020)
- IEEE ISSCC 270–271 (2022)

- Nature 608, 504–512 (2022)
- Nature Electronics 4, 635-644 (2021)
- Nature Electronics 2, 290-299 (2019)

The authors must cite all of these papers, not just from the context of CMOS and NVM collaboration, but also to provide a broader context of neuromorphic computing at the system level to position their paper accordingly. For the latter broader context, the authors should also cite the following two papers:

- Nature Electronics 1, 22–29 (2018).
- Nature Rev. Mater. 5, 173–195 (2020).

Overall, the authors have too narrowly assumed as if NVM devices were solutions to all problems, without capturing what it really takes to implement the entire neuromorphic DNN computing. The paper needs a major revision to set the tone and context correctly.

2. Equally importantly, what I stated in the foregoing bullet (1) then applies to their overclaim on “all-spintronic neuromorphic computing hardware”. Due to the necessity of CMOS electronics in building the entire DNN computing, implementing just NVMs and activation functions with spintronic devices cannot make the entire neuromorphic computing hardware spintronic. Even within the limited scope of the spintronic synapses and spintronic neurons of the authors, the authors should consider how spintronic synapses whose outputs are currents or voltages (depending on how the synaptic devices are arranged in a MAC column) will be interfaced with spintronic neurons whose inputs are the magnetic field, H_z . That is: can they immediately create H_z for spintronic neurons, from the current sum of the spintronic synapses? If that’s the case, are they strong enough? (and if that’s the case, how quantitatively current is translated into magnetic field based on this work?) Or does the interface require some intervention by CMOS electronic? The authors should also note that CMOS electronics for data converters (ADCs) will be absolutely needed either between synapses and neurons, or at least at the end of neurons, or likely at both positions. In sum, the fact that the authors demonstrated synapses and neurons with spintronic devices does not mean at all that the entire neuromorphic computing hardware can be built purely based on spintronics. The authors need to make a substantial revision to set the context properly.

3. The second and third sentences of the second paragraph of Introduction are also problematic (by the way, these sentences missed out flash memory and MRAM, which are commercially much more successful than, for example, PRAM). Different memory types have their own advantages and disadvantages: some are slow, some do not last long, some have larger areas, and some have only 1-bit operation. I do not believe that the spintronic devices the authors put forth here will win in all categories, so the above-mentioned sentences are exaggerations that unconditionally promote their own work. Much more scholarly balance is required.

4. When they use their device as a synapse with a proper arrangement of pinning centers and domain walls, they are memory elements. Now when they use their device as a neuron (an activation function)

with another suitable arrangement of pinning centers and domain walls, does it cease to be a memory? In the usual operation, the activation function is not supposed to have a memory behavior. The authors should clearly articulate whether the neuron (activation function) is a memory or not, how it is so, and if it is still a memory, whether it will be problematic or not in the scheme of DNN computing.

5. Also can the authors more clearly and easily explain how the nonlinearity arises to make the device perform as what generates the activation function?

6. If the authors created H_z with a current, they do need to specify the amount of current that is needed (I already mentioned this earlier). I had difficulties in finding this information, although the authors might have put it somewhere.

7. The paper feels a bit too long, and I think many technical details can go to supplementary information. Also the abstract is way too detailed to be broadly accessed. The authors should try to write this in a more accessible manner.

8. I believe Figures 4 and 5 are experimental highlights, and but then the paper has Figure 7, which is totally based on simulation, but with a very misleading caption of "Circuit-level implementation". While whether the authors leave Figure 7 in the paper or not is really up to their choice, I personally believe that this has an anti-climatic effect that draws the reader's attention from what they should focus on (the two key experimental demonstrations for synapses and neurons). But at the least, the authors must be very explicit that Figure 7 is purely based on simulation; while they do state that, it is not immediately obvious with the presently ambiguous descriptions. Certainly, wordings like "Circuit-level implementation" must not be used: they are certainly too misleading.

Response to Referees Letter

Manuscript ID: NCOMMS-23-45616

Title: Domain wall magnetic tunnel junction-based artificial synapses and neurons for all-spin neuromorphic hardware

Color code: Reviewer comments are in **bold**; and **blue** color is used for highlighting the changes in the revised manuscript and supporting information.

REVIEWERS' COMMENTS

Reviewer #1 (Remarks to the Author):

This paper reports an interesting endeavor to demonstrate a reliable, multi-state synaptic device based on domain wall MTJ and a sigmoid activation function generator. The concept, apart from the function generator scheme, is not new, but this work can be considered one of the first experimental reports in which the programmability and the distinction of states are adequately emphasized. While the experimental part on the device is interesting, the characterization of the circuit, or even a smaller sub-circuit containing the all spin device, along with CMOS part, is lacking and in my opinion that can be the differentiating factor that elevates research of this magnitude to the level of novelty that warrants publication in a prestigious journal like nature communication (more about this in my detailed comments).

Here are my most important concerns about this paper:

Our response:

We sincerely thank the reviewer for appreciating our work that can be considered **one of the first experimental reports in which the programmability and the distinction of states are adequately emphasized**. Based on the reviewer's comments on experimental part of a typical sub-circuit containing the all-spin devices along with CMOS part, we have carried out systematic experiments with data corroboration and made substantial revisions on our manuscript. We hope that the revisions addressed all concerns clearly. First, the brief responses are categorized as below, followed by our point-by-point replies.

Construction and characterization of the circuit: We have included additional construction and characterization of the circuit, specifically focusing on a representative sub-circuit that contains the all-spin synapse and neuron devices along with the CMOS part. This new characterization provides a more comprehensive understanding of the device performance and its integration with the CMOS circuitry. The experimental results and analysis have been included in the revised manuscript, highlighting the performance and interaction of the all-spin device and the CMOS components.

Novelty and distinction of states: We have further emphasized the novelty and distinction of states in our revised manuscript, with additional analysis and discussions to highlight the significance of our experimental results. These additions aim to emphasize the programmability and distinction of states, which are key aspects of our work as appreciated by reviewer.

We believe that the essential results were presented more effectively and were closely linked with the conclusions. Below we quoted the reviewer's comments in **bold**, followed by our point-by-point replies.

1. The circuit part of this paper is lacking in my opinion and needs a full revisit. First of all, since this is just a simulation, why did the authors limit the array to 4x4? By using a larger array and a longer pre-neuron pulse train (they can be closer to each other... why are the pulses have such a long separation?) they can showcase a larger range of states.

Our response:

We appreciate reviewer's comments regarding the circuit part of our paper. We understand your concern about the limitations of the 4x4 array in our simulation and the pulse train separation in the original manuscript. We agree that it is important to provide a comprehensive analysis of the capabilities of our developed all-spin neuromorphic computing hardware.

The choice of a 4x4 array and the separation of the pre-neuron pulse train in our simulation was primarily intended to provide a simple and effective demonstration for the feasibility of the proposed neuron device to transmit the nonlinear resistance state distribution defined by pinning center distance to nonlinear electrical output function. While a larger array and a longer pulse train could indeed showcase a larger range of states, our focus in this work was to establish the feasibility of all-spin synapses and sigmoidal activation function generators using DW-MTJs.

By employing a smaller array and a conservative pulse train separation, we aimed to highlight the basic functionalities and characteristics of our developed hardware, such as programmable multi-state synaptic devices, high reliability, and energy efficiency. However, we appreciate reviewer's suggestion to explore larger arrays and longer pulse trains to showcase a broader range of states.

To follow reviewer's valuable advice, in the revised manuscript, we have included the **updated Fig. 7** that now demonstrates a larger spin-synapses array, up to 16x16, with respective 16-state sigmoid neurons, as demonstrated below **Fig. 7a**. Additionally, we have extended the duration of the pulse train to 1 ns with a pulse width of 500 ps (**Fig. 7b**) to provide a more comprehensive understanding of the potential capabilities of our proposed hardware.

Updated Fig. 7 Circuit simulation and experimental verification of spin neuron circuit. **a** Schematic of a simplified ANN network comprising 16x16 DW-MTJ synapses and 16x1 16-state DW-MTJ neurons. **b** Transient simulation results of the spin neuron circuit.

Furthermore, we have conducted complementary micromagnetic simulations to verify the ability to

realize a larger range of states in the developed DW-MTJ spin-sigmoid neurons. We have included the corresponding results in the **updated Fig. S7** of the revised *Supporting Information*. These simulations support our findings and provide further evidence of the potential of our DW-MTJs in achieving multi-state functionality in a programmable non-linear resistance states manner.

Updated Fig. S7 Prototype sigmoidal activation function generator. **a** Schematic diagram of proposed spintronic neuron based on DW-pMTJ. **b** DW energy distribution as a function of DW position with different i DMI according to q - ϕ model. Regions in pink rectangles denote PCs with larger DMI constant. Micromagnetic simulation results: **c** DW energy distribution as a function of DW position, and **d** DW position under a sequential SOT current pulse. Blue rectangles refer to PCs.

By incorporating these micromagnetic simulations, we aim to strengthen the validation of our developed hardware and provide a more complete understanding of its capabilities. We believe that these additional results enhance the scientific rigor and completeness of our work.

Changes highlighted in the revised manuscript and Supporting Information:

We updated **Fig. 7** and **Fig. S7** in the revised manuscript and *Supporting Information*, respectively. The corresponding added descriptions are presented in the revised manuscript, page 17~19 in the section of **“Simulation and experimental verification of spin neuron circuit.** In this section, we present a simulation of a hybrid MTJ/CMOS system that incorporates a 16×16 DW-MTJ synapses and a 16×1 16-state DW-MTJ neurons. Additionally, a hardware implementation of one representative array in the simulation is corroborated in **Fig. 7.”**

We hope these revisions clearly address reviewer’s concerns and provide a more thorough exploration of the capabilities of our hardware implementation. Thank you for your constructive comments, and we appreciate your valuable inputs in improving the quality of our manuscript.

2. What is the purpose of Clock signal shown in fig 7? where is it applied? There's no indication anywhere.

Our response:

Thank you very much for pointing out the missing information. We acknowledge that the original Fig. 7 did not include the representation of the *Clock signal* and the *transistor*. We appreciate your attention to detail with valuable inputs.

The *Clock signal* serves the purpose of controlling the timing of write and read operations of the spin-neuron devices in our designed circuit. Specifically, it is applied to the gate of the transistor located between the input node of the inverter and the bottom electrode of the neuron devices. By controlling the timing of the *Clock signal*, we can synchronize the operations of the neuron devices and ensure proper functionality. The transistor is activated on the rising edge of the clock to perform read operations.

Changes highlighted in the revised manuscript:

We have taken this feedback into consideration and have **updated Fig. 7** in the revised manuscript to include the *Clock signal* and the *transistor*, providing a clearer visualization of the circuit's operation.

Updated Fig. 7a Schematic of a simplified ANN network comprising 16×16 DW-MTJ synapses and 16×1 16-state DW-MTJ neurons.

3. I suppose the Vpost-neuron signal is the node after the inverter, in fig 7a it's not properly placed, implying that it's before the inverter.

Our response:

Thank you very much for bringing up the issue regarding the placement of legend " $V_{\text{post-neuron } x}$ " in original Fig. 7a. We acknowledge that it was not properly placed and could have led to confusion.

To follow your advice, as illustrated in the **updated Fig. 7**, we have made the necessary adjustments in the **revised Fig. 7** as illustrated below to ensure that the legend " $V_{\text{post-neuron } x}$ " (abbreviated as " $V_{\text{post } x}$ ") is placed at the appropriate positions behind the output node of the inverter. By doing so, we aim to provide a clearer representation of the output of the inverter and avoid any potential misunderstandings.

Changes highlighted in the revised manuscript:

We have taken this feedback into consideration and have **updated Fig. 7** in the revised manuscript to ensure the legend " $V_{\text{post } x}$ " is placed at the appropriate positions.

Updated Fig. 7a Schematic of a simplified ANN network comprising 16×16 DW-MTJ synapses and 16×1 16-state DW-MTJ neurons.

4. A major problem with the circuit in 7a: If post neuron signal is captured after the inverter, as stated clearly in the manuscript, the inverter is supposed to give "full-swing" output, why do we see non full-swing signals in the last panel of fig 7b? (post neuron 3 and 4, as egamples) This is pretty confusing.

Our response:

Thank you for raising your concern about the behavior of the post-neuron signals in the bottom panel of original **Fig. 7b**. We appreciate your attention to detail and apologize for any confusion caused.

To address your concern, we would like to provide the following clarifications:

The description of "*full swing*" in the context of the inverter is indeed inappropriate. The primary objective of the original manuscript was to convey the comprehensive output range of the sigmoid function. In a digital logic circuit, "full swing" refers to the output voltage range where a high voltage represents a logic 1 (V_{DD}) and a low voltage represents a logic 0 (0 V)¹. However, in the case of the multi-level spin sigmoid activation function generator in our work, the output is an analog signal rather than a full swing logic level.

For post-neuron 1, the initial state is the antiparallel state (R_{AP}), where the output of the inverter is close to 0 V. After two cycles, post-neuron 1 transitions to the parallel state (R_P), and the output of the inverter is close to V_{DD} , exhibiting a swing closer to the full swing characteristic. However, for the case of post-neuron 3 and 4 in the **original Fig. 7b**, the configuration transitions to intermediate states in final, resulting in the voltage difference between final and initial states is not “full swing”.

This type of read scheme comprised of voltage divider network and inverter is widely used in spintronic neural network, most of which are spiking or binary neuron²⁻⁴. For our multi-state neuron, the function of the inverter is more similar to the PMOS utilized in Supplementary Ref. 5, to amplify the nonlinear signal and transmit it to the next layer. The read circuit in the present work is just a simple demonstration of feasibility of the developed DW-MTJ spin neuron device to realize multi-level electrical output in a sigmoidal profile. The read circuit and interconnection circuit may be much more complex with high precision ADC/DAC and operational amplifier in practical application, which is beyond the scope of this work.

Updated Fig. 7b. Transient simulation results of the spin neuron circuit. **Note:** “Post-neuron 4” in the original Fig. 7b changes into “Post-neuron 16” in the updated Fig. 7b for more clear visualization.

Specifically, we have made substantial revisions in the manuscript to address the issues raised. The following changes have been made: (1). We have revised the descriptions of the output behaviors in **Fig. 7b** to avoid the use of “full swing” terminology and provided a more accurate explanation of the behavior of the post-neuron signals. This revision helps clarify that the output of the inverter for post-neurons 3 and 4 (annotated as “16” in the updated Figs. 7a and 7b) represents intermediate analog values. (2). We have updated the corresponding text in the manuscript to reflect the revised explanation of the output behaviors. This ensures that the description aligns with the revised figure and accurately represents the behaviors of the circuit.

We appreciate your valuable input and constructive feedback. Thank you for bringing these concerns to our attention, and we apologize for any confusion caused by the initial description.

Changes highlighted in the revised manuscript:

As indicated in the revised manuscript, page 18 and 19, “Next, the resistance state is read out by the node voltage generated by the referencing circuit and transferred to a full-swing voltage output, labeled as post-neuron signal, by the inverter.” changes into “Next, the resistance state is read out by the voltage divider and amplified by the inverter to get the neuron output, labeled as post-neuron signal.”

5. The authors say "The output voltage with an evident “S” shape indicates that the developed sigmoidal multi-states are successfully transformed into circuit timing. " What do you mean by "s" shape output? I think there's some mistranslation happening here.

Our response:

Thank you for raising the question regarding the mention of an "S" shape output in the original manuscript. We apologize for any confusion caused by the original statement.

Updated Fig. 7b Transient simulation results of the DW-MTJ based circuit.

To clarify, in the context of the **original Fig. 7**, the reference to an "S" shape output pertains specifically to post-neuron 1. In this case, when all synapses connected to post-neuron 1 are in a low resistance state (weight "1"), state switching occurs after each pre-neuron signal is applied and reaches saturation within 7 clocks. The output voltage of post-neuron 1 exhibits a non-linear function of pulse number by time, which can be visualized as an "S" shape curve, as indicated by the black dash line in **updated Fig. 7b**.

However, for post-neurons 2, 3, and 4, some synapses connected to them are in a high resistance state (weight "0"). As a result, the pre-neuron signals weighted by these synapses do not drive state

switching in the corresponding post-neurons. Therefore, the output voltage of these post-neurons after 7 cycles does not exhibit an evident "S" shape with respect to pulse number by time.

The "S" shape is just an intuitive way to present data, which indicates that the state-by-state switching of the neuron device manipulated by pulse number follows the sigmoid function. However, in practical application, the output and impact on the subsequent level are solely determined by the final state. In response to reviewer's comments, we have made updates to **Fig. 7** in the revised manuscript. We have also included more specific discussions to address your concern.

Changes highlighted in the revised manuscript:

In page 17, as indicated in the revised manuscript, "The output voltage with an evident "S" shape indicates that the developed sigmoidal multi-states are successfully transformed into circuit timing." changes into "As a result, the output voltage of post-neuron1 is a non-linear function of pulse number by time in a sigmoidal profile, as indicated by the black dashed guideline."

6. The writing of "spin-neuron circuit implementation ..." section must be improved, it becomes unreadable at some points.

Our response:

We appreciate the reviewer's suggestions for improving the manuscript. Based on the reviewer's comments, we have made substantial revisions to the manuscript to address the concerns raised.

Changes highlighted in the revised manuscript:

We have taken this feedback into consideration and have **updated Fig. 7** in the revised manuscript. The subject of this section "**Spin-neuron circuit implementation leveraged by DW-MTJs synaptic and sigmoidal hardware**" changes into "**Simulation and experimental verification of spin neuron circuit.**"

We have made substantial revisions on this section according to the **updated Fig. 7** in the revised manuscript: "A simulation of a hybrid MTJ/CMOS system that incorporated 16×16 DW-MTJ synapses and a 16×1 16-state DW-MTJ neurons was performed (parameters listed in Table S2). Furthermore, to ensure the reliability of developed devices, we conducted a validation of the system's performance using a neuromorphic hardware implementation of representative arrays. The results of both the simulation and experiments are presented in **Fig. 7**. The purpose of such complementary study is to assess the feasibility of employing all-spin synapses and a sigmoidal activation function generator, as well as to evaluate the compatibility of our developed devices with CMOS technology. **Fig. 7a** illustrates the simplified artificial neural network (ANN) used in the simulation. Each synapse cell consists of a DW-MTJ-based synapse and two control transistors for write and read control. The gates of these transistors are connected to the word line (WWL) and the pre-neuron signal (V_{Pre}), respectively. The neuron cell comprises a DW-MTJ-based activation function generator, which forms a voltage divider with a reference resistor. It also includes a read control transistor and an inverter for output generation. The DW-MTJ-based devices are modeled using Verilog-A, and Supplementary Note S3 provides details on the co-simulation with the CMOS peripheral circuit. For simplicity, the DW-MTJ-based synapses operate in a binary mode with a default weight value.

Updated Fig. 7 Circuit simulation and experimental verification of spin neuron circuit. a Schematic of a simplified ANN network comprising 16×16 DW-MTJ synapses and 16×1 16-state DW-MTJ neurons. **b** Transient simulation results of the spin neuron circuit. **c** Hardware implementation of with developed 4 binary SOT-MTJs as synapse and 1 spin-neuron device (right panel). **d** Waveform of the pre-neuron signal output by MCU and corresponding output of the operational amplifier. **e** MOKE Kerr images captured after each pre-neuron signal applied.

The transient simulation results presented in **Fig. 7b** provide insights into circuit-level neuromorphic behaviors. Pre-neuron signal is applied asynchronously to the corresponding column to open all read control transistors in this column. The read current from source line (SL) flowing through DW-MTJ is sent to the operational amplifier generating a weighted current pulse injected into the write channel of the activation function generator to drive the DW motion and state refreshing. Next, the resistance state is read out by the voltage divider controlled by clock signal and amplified by the inverter to get the neuron output, labeled as V_{Post} . Since all the synapses connected to activation function generator 1 are in low resistance state, the DW in this generator is driven to next PC in each cycles. The output voltage of post-neuron1 is a non-linear function of pulse number by time in a sigmoidal profile, as indicated by the black dashed guideline.

The experimental construction and characterization of the circuit were carried out, specifically focusing on a representative sub-circuit that contains the all-spin synapse and neuron devices along with the CMOS part as illustrated in **Figs. 7c-7e**. Such characterization provides a more

comprehensive understanding of the device performance and its integration with the CMOS circuitry. The hardware setup (**Fig. 7c**) consists of one partial array with four binary SOT-MTJs serving as synapses and one proposed neuron device. The synapses are wire-bonded to a PCB carrier, which is connected to a universal board containing an MCU and peripheral circuits. The pre-neuron signals generated by the MCU are measured along with the output of the operational amplifier using an oscilloscope (**Fig. 7d**).

The experiment includes a sub-circuit of the array with CMOS components, resistors, and the actual device, allowing for the creation of a sample post-neuron. Supplementary Video 1 provides further detailed correlation. The resulting Kerr images (**Fig. 7e**) show the progression of the spin-neuron device. A down domain is initially nucleated at the left pad and propelled into the racetrack by current pulses triggered by the pre-neuron signals. The down-up DW is pinned at the corresponding pinning center after each cycle of the pre-neuron signal, causing a switch in the resistance state. This hardware test effectively demonstrates the functional characteristics, integration features, and implementation effects of the developed DW-MTJ all-spin synapses and neurons.”

7. The calculation (simulation?) results related to power/energy consumption are not clearly shown in the paper and any discussion on energy is limited to Fig s7 in the supplementary. the reported 3.5fj energy is per pulse, but an "operation" does not consist of a single pulse, but multiple pulses. So it must be clarified what the authors mean by "operation" when they report 508 fj/operation. I suggest they add a section in supplementary material explaining how they came up with this number. It's important to consider the difference in the meaning of "operation" in this case in which the device must be "settled" to its desired state, versus a simple logic like an inverter which flips between two definite states. For example, if as mentioned in the paper, it takes 7 cycles for DW to reach the last PC, we are dealing with a very different type of "operation" and the actual energy consumption can only be calculated by averaging for all combinations of input/outputs and settling times.

Our response:

Thank you for your insightful question regarding the calculation and presentation of power/energy consumption. We appreciate your valuable feedback and we understand the need for clarification on this aspect.

In response to the question and concern, we have provided more extensive explanations in both revised manuscript and *Supporting Information*. The reported energy consumption values, such as the updated 36.3 fJ per pulse and 508 fJ per operation, are based on simulations performed using the mumax³, an open-source GPU-accelerated micromagnetic simulation program software.

As for the 8-state neuron device in the original **Fig. 7** or **Fig. S7**, seven write pulses and a reset operation was included for the energy consumption estimation. The same as the 14-state neuron case, thirteen write pulses and a reset operation was included. As for the 14-state neuron device, the specific value is calculated based on the following corroborated equations as utilized in previous reports^{6,7}:

$$\begin{aligned}
 P &= I^2R = (J_e wh)^2 \frac{\rho l}{wh} = J_e^2 wh \rho l \\
 &= (0.55 \times 10^{12} \text{ A/m}^2)^2 \times 50 \times 10^{-9} \text{ m} \times 5 \times 10^{-9} \text{ m} \times 2 \times 10^{-6} \text{ m} \times 230 \\
 &\quad \times 10^{-8} \Omega \cdot \text{m} = 347.875 \text{ } \mu\text{W}
 \end{aligned}$$

$$E = (P \times t_p) \times n = 347.875 \times 10^{-6} \text{ W} \times 500 \times 10^{-12} \text{ s} \times 26 = 4.52 \text{ pJ/Operation}$$

where w , h , ρ , l , corresponds to width, thickness, resistivity and length of SOC layer, respectively. t_p refers to pulse width of write pulse. n is the total pulse number in an operation comprising of set and reset processes. Both of the set and reset processes are implemented by state-by-state DW motion driven by a series of identical pulse. Hence, for a 14-state device: $n = 2 \times 13 = 26$. Certainly, to reduce latency in this case, reset can be achieved by increasing the current density or pulse amplitude, which in turn increases the energy consumption.

To address reviewer's suggestion, we have added a comprehensive section of **Note S4: Performance evaluation of sigmoid activation function generator** in the revised *Supporting Information* that explains in detail how we arrived at the numbers for power/energy consumption. This section provides a thorough explanation of the calculations, taking into consideration factors such as *settling times, different combinations of inputs and outputs, and the specific characteristics* of the devices.

By including this additional section in the revised *Supporting Information*, we aim to provide a more comprehensive and transparent description of the power/energy consumption calculations, addressing the difference in the meaning of an "operation" in the context of our device compared to a simple logic operation like an inverter^{8,9}.

Further details regarding the calculation and presentation of power/energy consumption in our revised paper are as follows:

Energy Consumption per Pulse: The reported energy consumption of 36.3 fJ per pulse pertains to the energy required to switch one state by one single pulse in the device. This value is obtained through simulations using the Mumax³ software. It represents the energy dissipated during the switching of the magnetic states in the device.

Energy Consumption per Operation: The reported energy consumption of 508 fJ per operation considers a full swing/complete operation cycle scenario for the device. In the context of the 8-state neuron device depicted in **updated Fig. 7**, an "operation" consists of seven write cycles followed by a reset operation. These operations are necessary to achieve the desired state in the device. The energy consumption per operation is calculated by considering the energy dissipation during these write cycles and the reset operation.

Consideration of Operation Times: We acknowledge the importance of operation times in the calculation of energy consumption. In our case, where it takes seven cycles for the domain wall (DW) to reach the last pinning center (PC), operation times are considered in the energy calculations. We consider different combinations of inputs and outputs, as well as the time required for the DW to reach the desired state, in order to accurately estimate the energy consumed during the operation.

It is a tradeoff between recognition accuracy of the final network and energy consumption. However, recognition accuracy is fast to saturation with the increasing bit number of the sigmoid activation function generator. In most case, a 3-bit (8-state) sigmoid activation function generator is enough for accuracy^{10,11}. Hence, the energy/power consumption values of 8-state scheme was presented in the revised manuscript.

$$\begin{aligned}
P &= I^2 R = (J_e w h)^2 \frac{\rho l}{w h} = J_e^2 w h \rho l \\
&= (0.33 \times 10^{12} \text{ A/m}^2)^2 \times 50 \times 10^{-9} \text{ m} \times 5 \times 10^{-9} \text{ m} \times 1 \times 10^{-6} \text{ m} \times 230 \\
&\quad \times 10^{-8} \text{ } \Omega \cdot \text{m} = 62.6 \text{ } \mu\text{W} \\
E &= P \times t_p = 62.6 \times 10^{-6} \text{ W} \times 580 \times 10^{-12} \text{ s} = 36.3 \text{ fJ/Pulse} \\
E &= (P \times t_p) \times n = 62.6 \times 10^{-6} \text{ W} \times 580 \times 10^{-12} \text{ s} \times 14 = 508 \text{ fJ/operation} \\
f &= \frac{1}{n \times t_p} = \frac{1}{14 \times 580 \times 10^{-12} \text{ s}} = 20 \text{ MHz}
\end{aligned}$$

Updated Fig. S9. Benchmark of latest representative sigmoidal activation function generator¹¹⁻¹⁷.

To follow reviewer's advice, we have added a comprehensive section of Note S4: Performance evaluation of sigmoid activation function generator to the *Supplementary Material* that provides a detailed explanation of how we arrived at the power/energy consumption values. This section includes the relevant equations and factors considered in the calculations, ensuring transparency and clarity in our reporting.

Compared to state-of-the-art counterparts of neural activation function generator, our developed architecture is competitive as shown in **Fig. S9** benchmark diagram and below tabulated benchmarking features in Supplementary Table 1. The overall energy consumption is less than **508 fJ/operation** including the reset process with a competitive firing rate up to 20 MHz.

We hope these additional information provides further insights into the calculation and reporting of power/energy consumption in our paper. Thank you for bringing up this important point, and we appreciate your feedback.

Changes highlighted in the revised Supporting Information:

To follow the referee's advice, we added a session Note S4 with specific discussions upon systematic devices' performance evaluation. Please refer to the *Supporting Information* for more details.

Supplementary Table 1. Benchmark of latest representative sigmoidal activation function generator from device structure, investigation methodology and Energy/Power consumption perspectives¹¹⁻¹⁹.

	Discrete DW-MTJ ¹²	DW-hall bar ¹³	DW-hall bar ¹⁸	DW- LUT ²⁹	SOT- MTJ ¹⁴	Skyrmion- MTJ ¹¹	180 nm CMOS ¹⁵	28 nm CMOS ¹⁶	180 nm CMOS ¹⁷	This work		
Investigation methodology	Exp.	Exp.	Exp.	Sim. (8-bits)	Sim.	Sim. (3-bits)	Sim.	Sim.	Sim.	Exp. (8-States)	Sim. (8/14-states)	
Area (μm^2)	10.5	>15600	>120	31.8	0.138	>0.63	89250	680.5	38.556	150	0.05/0.1	
Operation Period	8 ns	500 ms	5 us	2 cycles	—	2.5 ns	—	—	—	280 us	50.4/104 ns	
Energy consumption	<16 pJ	~0.5 J	—	116 pJ	—	4.1 fJ	—	—	—	Per Pulse	1.21 μJ	36.3/173.9 fJ
										Per Operation	16.9 μJ	508/4520 fJ
Power consumption	0.15-2 mW	~1 W	—	—	18.04 μW	5.67 μW	8.02 μW	493.4 μW	62.5 μW	0.121 W	62.6/347.9 μW	

8. I strongly suggest that the authors setup a partial experiment and include part of the array, it can be just top row with 4 CMOS, 4 low/high resistors and the actual device, creating a sample post neuron. For me that experiment qualifies this paper for publication.

Our response:

Thank you for your valuable suggestion. We appreciate your advice with verification of experiment that includes the actual device, corroborating a sample post neuron as part of the array. We understand the importance of experimental validation in scientific research, and we recognize that conducting such experiments can provide further support for the findings presented in our paper.

Supplementary Fig. 1. Schematic diagram of the experiment.

To address your suggestion, we conducted extensive and supplementary experiments to validate our research. Specifically, we investigated the current state-of-the-art in hardware integration of non-volatile memory-based artificial synapses and neurons with MCU-FPGA-PCB brain-inspired computing demonstrations^{20,21}. Based on established processes and technical approaches, we designed and implemented our own sub-circuit implementation demonstration using DW-MTJs all-spin synapses and neurons.

To optimize the hardware integration, we designed a tailored hardware integration scheme that considered the device characteristics of the developed DW-MTJs all-spin synapses and neurons, as well as the available laboratory platform resources. **Supplementary Fig. 1a** is a schematic diagram of the whole experiment system, in which a Labview program in host computer (HC) is used to control timing sequence. First, command sent from HC to MCU on board (STC89C52RC) triggers pre-neuron signals (pulse width 1.5 s) at the I/O ports connected to 5 V power supply through pull-up resistors. The pre-neuron signals are applied to switch on the transistors (2N7000) connected in series with SOT-MTJ synapses in the peripheral circuits as shown in **Supplementary Fig. 1b**. The SOT-MTJ synapses are connected in series with a reference resistor with intermediate resistance approximately of 426 Ω $[(R_{AP} + R_P)/2]$ constituting a voltage divider with read voltage V_{Read} of 30 mV. The node voltage is fed into operational amplifier (LM741CN) with bipolar power supply (V_{cc} 5 V, V_{ee} -5 V), and compared to a reference voltage V_{Ref} of 16 mV subsequently, to generate a output pulse consequently to switch on the transistor (2N7000) linking current pulse generator (Keithley 6221) and neuron device. With appropriate delay, a pulse command is sent to Keithley 6221 to trigger

a current pulse (20 μ s in pulse width) applied between SOT channel to switch state of neuron device. The MOKE system, synchronously controlled by the Labview program through communication with MagVision interface starts to capture Kerr images after the current pulse applied. **Supplementary Fig. 1c** depicts the schematic diagram of designed timing sequence of aforementioned operations. In fact, the modules are not strictly synchronized. Therefore, a prolonged pre-neuron signal (1.5 s) is utilized for effective demonstration on purpose.

Supplementary Fig. 2. Schematic simulation of the designed circuit by Proteus ISIS.

Upon complementary execution and verification, we simulated the designed circuit schematic using Proteus ISIS before the experimental implementation, as shown in **Supplementary Fig. 2**. The simulation and debugging results included the MCU model (STC89C52RC) and specific outputs such as serial port control, microcontroller simulation, and circuit output.

The actual experiment setup is demonstrated in **Supplementary Fig. 3a**, with every module labeled by a white tag. The core component, a universal board with MCU and peripheral circuits, is marked by orange dashed box in **Supplementary Fig. 3a**, the magnified view of which is shown in **Supplementary Fig. 3b**. Four SOT-MTJs are connected to a PCB carrier by wire bonding to serve as synapses, highlighted by purple dashed box in **Supplementary Fig. 3b**. The PCB carrier is then attached to another universal board by jump wire, which is connected to other peripheral circuits by pin headers. A typical SOT switching loop of the SOT-MTJ synapse is shown **Supplementary Fig. 3c**, with R_p , R_{AP} of 296, 557 Ω respectively, corresponding to a TMR of 88%. All SOT-MTJ synapses were set to R_{AP} state at begin by SOT current pulse with 5 μ s pulse width and 15 mA pulse amplitude.

Supplementary Fig. 3. **a** Photo of the whole experiment system. **b** Hardware implementation of one partial array in Fig. 7 a with 4 binary SOT-MTJ as synapse and 1 proposed neuron device. **c** R-I loop of the SOT-MTJ used as synapse in **b**.

The measured results are demonstrated in **Supplementary Fig. 4**, where Figs. 4a-4d refer to pre-neuron signals from MCU and Fig. 4e refers to output of the operational amplifier measured by the oscilloscope. As all SOT-MTJs synapses were in R_{AP} state, output of the operational amplifier was +1 (+4 V) when each pre-neuron signal turned on to enable a write pulse from Keithley 6221 accordingly. While output of the operational amplifier was -1 (-2.5 V) to cut off the write path when pre-neuron signal turned off. The resulted Kerr images of the neuron device with 7 pinning centers marked by red boxes is shown in **Supplementary Fig. 4f**. A down domain (dark region) was nucleated firstly at the left pad, which was pushed into the racetrack by a series of current pulses triggered by the pre-neuron signals shown in **Supplementary Fig. 1c** in sequence then. The down to up domain wall arrived and got pinned at corresponding pinning center (PC) after each pre-neuron signal cycle leading to resistance state switch according to the equation:

$$R_{MTJ} = R_P \left(\frac{x_0}{L} \right) + R_{AP} \left(1 - \frac{x_0}{L} \right)$$

where x and L refer to the domain wall position, length of MTJ respectively.

The hardware test results effectively demonstrate the functional characteristics, integration features, and implementation effects of the developed DW-MTJs all-spin synapses and neurons. We created a systematic and detailed experiment that included part of the array sub-circuit, consisting of 4 CMOS components, 4 SOT-MTJ synapses, and the actual device. This configuration allowed us to create a sample post neuron. **Supplementary Video 1** provides further correlation in detail.

Supplementary Fig. 4. a-e Waveform of the pre-neuron signal output by MCU and corresponding output of operational amplifier. f MOKE Kerr images captured after each pre-neuron signal applied.

We believe that the aforementioned experiment implementation is valid to sustain the main concepts of this work: feasibility of prototype device, potential for all-spin neural network, potential for compatibility with CMOS process and parts. Nevertheless, there are still two main limitations we have to emphasize:

First, binary SOT-MTJ was chosen to serve as synapse due to small junction resistance (\sim few Ω) of multistate synaptic device which is challenge to be differentiated by the voltage divider network with potentiometer (3296w 501) and the operational amplifier. This limitation could be solved by device size scaling down or increase (Resistance-Area-Product) RA of film stack (205.879 $\Omega\mu\text{m}^2$ of the present film stack) according the relation between magnetoresistance, area and RA²².

Supplementary Fig. 5. **a-c** Dilemma of simultaneous electrical write and detection in the present DW-MTJ based synapse and neuron. **d-e** Solution to above problem by device size shrinking and increasement of RA to increase junction resistance.

Second, the dilemma of simultaneous electrical write and detection in the multistate device, owing to non-trivial shunting effect of write current resulted by the central top electrode (Ti/Au 20nm/80 nm) which is crucial for multi-state read for large device size with small junction resistance (\sim few Ω), as shown in **Supplementary Figs. 5a-5c**. However, when the device scaling down to nano-scale or increase the RA as shown in **Supplementary Fig. 5d**, the magnetoresistance outweighs longitudinal resistance of write channel in device which eliminate the demand of central top electrode for read operation. As shown in **Supplementary Fig. 5e**, the read current injected from left top electrode flows along write channel and then flows down to bottom electrode at each section. While the write current mainly flows between left and right top electrode to maximum effective write current.

These two main limitations both are applicable to be settled by device size scaling. And the nano-scale device fabricated through electron beam lithography (EBL) process has been carried out with preliminary results, but requiring further optimization due to the limited process conditions in the current clean room.

Except that continuing optimization of our EBL process and nano-scale device performance, we have also carried out device fabrication of synapses and neurons crossbar arrays in a more applicable manner for integration with CMOS peripheral circuits, which is a preliminary exploration of future tape-out plan in standard foundry. The layout and the optical microscopy images of preliminary crossbar arrays is demonstrated in **Supplementary Fig. 6**. We have outlined the specific chip fabrication process to ensure CMOS process-compatible wafer-level system integration and architectural functionality demonstration.

Supplementary Fig. 6. The layout and optical images of the proposed all-spin DW-MTJs synapses and neurons cross bar array for planned tape out processing.

The preliminary experimental demonstration, device optimization analysis, and the extended crossbar array-level devices enrich our presentation and highlight the significance of this work with applicable methods. We believe these detailed illustrations will benefit future studies, and we sincerely thank the reviewer for motivating us to do this.

In the revised manuscript, we have included a comprehensive description of the experimental setup, methodology, and results obtained from this partial experiment. We have also provided a detailed analysis and comparison of the experimental data with the simulated results, further validating the performance and functionality of the proposed system.

We sincerely appreciate your suggestion and the emphasis you place on the importance of experimental verification. By incorporating this partial experiment into our research, we aim to strengthen the scientific rigor of our work and provide a more comprehensive assessment of the proposed approach.

We value your input and have endeavored to provide a more professional and scientific response. We believe that these additional experiments and optimizations enhance the overall research and contribute to the robustness and applicability of our proposed approach.

Changes highlighted in the revised manuscript:

Supporting Information Video S7 aims to visualize aforementioned experiment implementation, we added an extensive discussions in the section of **Simulation and experimental verification of spin neuron circuit**: “A simulation of a hybrid MTJ/CMOS system that incorporated 16×16 DW-MTJ synapses and a 16×1 16-state DW-MTJ neurons was performed (parameters listed in Table S2).

Furthermore, to ensure the reliability of developed devices, we conducted a validation of the system's performance using a neuromorphic hardware implementation of representative arrays. The results of both the simulation and experiments are presented in **Fig. 7**. The purpose of such complementary study is to assess the feasibility of employing all-spin synapses and a sigmoidal activation function generator, as well as to evaluate the compatibility of our developed devices with CMOS technology. **Fig. 7a** illustrates the simplified artificial neural network (ANN) used in the simulation. Each synapse cell consists of a DW-MTJ-based synapse and two control transistors for write and read control. The gates of these transistors are connected to the word line (WWL) and the pre-neuron signal (V_{Pre}), respectively. The neuron cell comprises a DW-MTJ-based activation function generator, which forms a voltage divider with a reference resistor. It also includes a read control transistor and an inverter for output generation. The DW-MTJ-based devices are modeled using Verilog-A, and Supplementary Note S3 provides details on the co-simulation with the CMOS peripheral circuit. For simplicity, the DW-MTJ-based synapses operate in a binary mode with a default weight value.

The transient simulation results presented in **Fig. 7b** provide insights into circuit-level neuromorphic behaviors. Pre-neuron signal is applied asynchronously to the corresponding column to open all read control transistors in this column. The read current from source line (SL) flowing through DW-MTJ is sent to the operational amplifier generating a weighted current pulse injected into the write channel of the activation function generator to drive the DW motion and state refreshing. Next, the resistance state is read out by the voltage divider controlled by clock signal and amplified by the inverter to get the neuron output, labeled as V_{Post} . Since all the synapses connected to activation function generator 1 are in low resistance state, the DW in this generator is driven to next PC in each cycles. The output voltage of post-neuron1 is a non-linear function of pulse number by time in a sigmoidal profile, as indicated by the black dashed guideline.

The experimental construction and characterization of the circuit were carried out, specifically focusing on a representative sub-circuit that contains the all-spin synapse and neuron devices along with the CMOS part as illustrated in **Figs. 7c-7e**. Such characterization provides a more comprehensive understanding of the device performance and its integration with the CMOS circuitry. The hardware setup (**Fig. 7c**) consists of one partial array with four binary SOT-MTJs serving as synapses and one proposed neuron device. The synapses are wire-bonded to a PCB carrier, which is connected to a universal board containing an MCU and peripheral circuits. The pre-neuron signals generated by the MCU are measured along with the output of the operational amplifier using an oscilloscope (**Fig. 7d**).

The experiment includes a sub-circuit of the array with CMOS components, resistors, and the actual device, allowing for the creation of a sample post-neuron. Supplementary Video 1 provides further detailed correlation. The resulting Kerr images (**Fig. 7e**) show the progression of the spin-neuron device. A down domain is initially nucleated at the left pad and propelled into the racetrack by current pulses triggered by the pre-neuron signals. The down-up DW is pinned at the corresponding pinning center after each cycle of the pre-neuron signal, causing a switch in the resistance state. This hardware test effectively demonstrates the functional characteristics, integration features, and implementation effects of the developed DW-MTJ all-spin synapses and neurons.”

In the section of **Circuit simulation and experimental validation (Methods)**, we added “As to the experiment system for spin-neuron sub-circuit implementation, a Labview program in host computer was used to control timing sequence with synchronizing all-spin NC hardware with integration of CMOS components (for more information refer to SI).”

Supplementary References

- 1 Chi, L. J., Yu, M. J., Chang, Y. H. & Hou, T. H. 1-V Full-Swing Depletion-Load a-InGaZnO Inverters for Back-End-of-Line Compatible 3D Integration. *IEEE Electron. Device. Lett.* **37**, 441-444 (2016).
- 2 Sharad, M., Fan, D. & Roy, K. Spin-neurons: A possible path to energy-efficient neuromorphic computers. *J. Appl. Phys.* **114**, 234906 (2013).
- 3 Sengupta, A., Choday, S. H., Kim, Y. & Roy, K. Spin orbit torque based electronic neuron. *Appl. Phys. Lett.* **106**, 143701 (2015).
- 4 Sengupta, A., Yogendra, K., Fan, D. & Roy, K. Prospects of efficient neural computing with arrays of magneto-metallic neurons and synapses. In *2016 21st Asia and South Pacific Design Automation Conference (ASP-DAC)*. 115-120.
- 5 Sengupta, A., Shim, Y. & Roy, K. Proposal for an All-Spin Artificial Neural Network: Emulating Neural and Synaptic Functionalities Through Domain Wall Motion in Ferromagnets. *IEEE Trans. Biomed. Circuits. Syst.* **10**, 1152-1160 (2016).
- 6 Hassan, N. *et al.* Magnetic domain wall neuron with lateral inhibition. *J. Appl. Phys.* **124**, 152127 (2018).
- 7 Tuma, T., Pantazi, A., Le Gallo, M., Sebastian, A. & Eleftheriou, E. Stochastic phase-change neurons. *Nat. Nanotechnol.* **11**, 693-699 (2016).
- 8 Adam, G. C., Khiat, A. & Prodromakis, T. Challenges hindering memristive neuromorphic hardware from going mainstream. *Nat. Commun.* **9**, 5267 (2018).
- 9 Liu, X. *et al.* Neuromorphic Nanoionics for human-machine Interaction: from Materials to Applications. *Adv. Mater.*, 2311472 (2024).
- 10 Ramasubramanian, S. G., Venkatesan, R., Sharad, M., Roy, K. & Raghunathan, A. in *Proceedings of the 2014 international symposium on Low power electronics and design* 15-20 (2014).
- 11 He, Z. & Fan, D. A tunable magnetic skyrmion neuron cluster for energy efficient artificial neural network. In *Design, Automation & Test in Europe Conference & Exhibition (DATE), 2017*. 350-355.
- 12 Siddiqui, S. A. *et al.* Magnetic Domain Wall Based Synaptic and Activation Function Generator for Neuromorphic Accelerators. *Nano Lett.* **20**, 1033-1040 (2020).
- 13 Yang, S. *et al.* Integrated neuromorphic computing networks by artificial spin synapses and spin neurons. *NPG Asia Mater.* **13**, 4057 (2021).
- 14 Amin, M. H., Elbtity, M., Mohammadi, M. & Zand, R. in *Proceedings of the Great Lakes Symposium on VLSI 2022* 319-323 (Association for Computing Machinery, Irvine, CA, USA 2022).
- 15 Xing, S. & Wu, C. Implementation of A Neuron Using Sigmoid Activation Function with CMOS. In *2020 IEEE 5th International Conference on Integrated Circuits and Microsystems (ICICM)*. 201-204.
- 16 Baccelli, G., Stathis, D., Hemani, A. & Martina, M. NACU: A Non-Linear Arithmetic Unit for Neural Networks. In *2020 57th ACM/IEEE Design Automation Conference (DAC)*. 1-6.
- 17 Shamsi, J. *et al.* Hyperbolic tangent passive resistive-type neuron. In *2015 IEEE International Symposium on Circuits and Systems (ISCAS)*. 581-584.
- 18 Zhou, J. *et al.* Spin-Orbit Torque-Induced Domain Nucleation for Neuromorphic Computing. *Adv. Mater.* **33**, e2103672 (2021).
- 19 Yu, H. *et al.* Energy efficient in-memory machine learning for data intensive image-processing by non-volatile domain-wall memory. In *2014 19th Asia and South Pacific Design Automation Conference (ASP-DAC)*. 191-196.
- 20 Park, J. *et al.* Implementation of Convolutional Neural Networks in Memristor Crossbar Arrays with Binary Activation and Weight Quantization. *ACS Appl. Mater. Interfaces* **16**, 1054-1065 (2024).
- 21 Kim, D.-W. *et al.* Real-Time Correlation Detection via Online Learning of a Spiking Neural Network with a Conductive-Bridge Neuron. *Adv. Electron. Mater.* **8**, 2101356 (2022).
- 22 Tsunekawa, K. *et al.* Giant tunneling magnetoresistance effect in low-resistance CoFeB/MgO(001)/CoFeB magnetic tunnel junctions for read-head applications. *Appl. Phys. Lett.* **87**, 072503 (2005).

Reviewer #2 (Remarks to the Author):

The major experimental contributions of this paper lie in acquiring both multi-state memory behaviors ("synapse") and activation function("neuron") by utilizing a spintronic device, in particular, by controlling the distribution of domain-wall pinning centers. I find the experiment and device concept themselves quite exciting. However, this excellent experimental advance is masked and compromised by many exaggerated, non-factual, or illogical statements such as "all-spintronic neuromorphic computing hardware" that appear throughout the manuscript, which are not consistent with, and show the misunderstanding of, how deep neural network (DNN) computing is actually done at the system level. For acceptance, the present form of the manuscript needs a substantial revision so that its logic and contents can be much more robustly structured just on the essence of the work with scientific rigor and scholarly balance.

Our response:

We are grateful to the reviewer's positive evaluation and favorable support. Based on the reviewer's comments, we have made substantial revisions on our manuscript. We believe that the essential results were presented more effectively and were closely linked with the conclusions. We hope that the revisions addressed all questions clearly. Below we quoted the reviewer's comments in **bold**, followed by our point-by-point replies.

1. An example of the illogical, non-factual, or exaggerated statement is the first sentence of the second paragraph of Introduction, which is contradictory to how the DNN computing is actually done. That is, NVMs and CMOS electronics are not what should be separated and compared against each other, but what should be used together for neuromorphic DNN computing. Concretely, a DNN consists of many layers, and each layer is a concatenation of the multiply-accumulate (MAC) operations for matrix multiplications, the activation function for decision making or thresholding, and the data conditioning and trimming such as pooling, max-pooling, soft-maxing, and many more. Thus the actual neuromorphic DNN implementation requires, for each layer, not just NVMs to store synaptic weights to perform the MAC operations, but also CMOS electronics to perform data thresholding (ok, in authors case, this may one day be possibly done with the same spintronic device in the "neuron" function), data conversion (necessary due to the analog computation with NVMs), data conditioning, and data trimming. So NVMs and CMOS electronics (mixed-signal and digital electronics) are not two competing technologies, but complementary technologies that should work together for neuromorphic DNN computing (and the industry already embeds NVMs such as RRAMs and MRAMs in commercial CMOS technology). For directly relevant examples as to how CMOS and NVMs must work together or how CMOS and various NVMs are available together through industry, please see

- **Nature 577, 641–646 (2020)**
- **IEEE ISSCC 270–271 (2022)**
- **Nature 608, 504–512 (2022)**
- **Nature Electronics 4, 635-644 (2021)**
- **Nature Electronics 2, 290-299 (2019)**

The authors must cite all of these papers, not just from the context of CMOS and NVM collaboration, but also to provide a broader context of neuromorphic computing at the system level to position their paper accordingly. For the latter broader context, the authors should also

cite the following two papers:

- **Nature Electronics 1, 22–29 (2018).**
- **Nature Rev. Mater. 5, 173–195 (2020).**

Overall, the authors have too narrowly assumed as if NVM devices were solutions to all problems, without capturing what it really takes to implement the entire neuromorphic DNN computing. The paper needs a major revision to set the tone and context correctly.

Our response:

We appreciate the reviewer’s advice and suggestions for improving our manuscript. Indeed, NVMs and CMOS electronics (mixed-signal and digital electronics) are not two competing technologies, but complementary technologies that should work together for neuromorphic DNN computing. Based on the reviewer's comments, we have made substantial revisions to the manuscript to address the concerns raised and provide a more comprehensive and accurate portrayal of the contributions and context of our work.

In the revised manuscript, we have revised the statements and now clearly emphasize the complementary nature of NVMs and CMOS electronics in neuromorphic DNN computing. We acknowledge that a DNN consists of multiple layers, each involving MAC operations, activation functions, and various data processing. We clarify that while our work focuses on the spintronic device for implementing the synapses and neuron activation function, we recognize that CMOS electronics are essential for other tasks such as data thresholding, conversion, conditioning, and trimming as advised by the reviewer.

To provide a broader context of neuromorphic computing at the system level and position our work more accurately, we have included all the references suggested by the reviewer. These references cover relevant works that discuss the collaboration between CMOS and various NVMs, as well as provide a comprehensive overview of neuromorphic computing. We have cited all these papers accordingly to support our statements and demonstrate the existing research in the field.

We believe that the revised manuscript now sets the tone and context more accurately. We appreciate the reviewer's valuable inputs, and we are confident that the revised manuscript provides a robust and scientifically rigorous account of our experimental contributions while maintaining scholarly balance.

Changes highlighted in the revised manuscript:

To follow the referee’s advice, first sentence of the second paragraph of Introduction has been revised to reflect the complementary nature of NVMs and CMOS electronics in neuromorphic DNN computing. “Notably, hardware implementation of neuromorphic devices based on emerging nonvolatile memories (NVMs) offers significant performance advantages when combined with traditional complementary metal-oxide semiconductor (CMOS) technology. The integration of NVMs and CMOS electronics provides benefits such as nonvolatility, scalability, direct mapping of synaptic weights, as well as facilitating functions such as data thresholding, conversion, and trimming required for each layer of a neuromorphic DNN^{7,13-18}.”

2. Equally importantly, what I stated in the foregoing bullet (1) then applies to their overclaim on “all-spintronic neuromorphic computing hardware”. Due to the necessity of CMOS electronics in building the entire DNN computing, implementing just NVMs and activation functions with spintronic devices cannot make the entire neuromorphic computing hardware spintronic. Even within the limited scope of the spintronic synapses and spintronic neurons of the authors, the authors should consider how spintronic synapses whose outputs are currents or voltages (depending on how the synaptic devices are arranged in a MAC column) will be interfaced with spintronic neurons whose inputs are the magnetic field, H_z . That is: can they immediately create H_z for spintronic neurons, from the current sum of the spintronic synapses? If that’s the case, are they strong enough? (and if that’s the case, how quantitatively current is translated into magnetic field based on this work?) Or does the interface require some intervention by CMOS electronic? The authors should also note that CMOS electronics for data converters (ADCs) will be absolutely needed either between synapses and neurons, or at least at the end of neurons, or likely at both positions. In sum, the fact that the authors demonstrated synapses and neurons with spintronic devices does not mean at all that the entire neuromorphic computing hardware can be built purely based on spintronics. The authors need to make a substantial revision to set the context properly.

Our response:

Thank you for highlighting the importance of properly setting the context and clarifying the concept of "all-spintronic neuromorphic computing hardware." We appreciate your insights and suggestions. The "all-spintronic neuromorphic computing hardware" concept is not new, but widely used in many other works in fact¹⁻⁵. As mentioned in Supplementary Ref. 1, “an all-spin neural network” is the one where “spintronic device offers a direct mapping to synapses and neuron functionalities in the brain while inter-layer network communication is accomplished via CMOS transistors”. In summary, the proposed all-spintronic neuromorphic computing hardware is comprised of spintronic devices serving as the core elements of neural network (synapses and neurons), and CMOS peripheral circuits for inter-layer connection, data conversion, clock control, *etc.* This definition totally coheres with your views. As you pointed out, spintronic devices and CMOS electronics are not two competing technologies, but complementary technologies to construct neuromorphic DNN computing.

We acknowledge that the term “all-spintronic neuromorphic computing hardware” may have been misleading or ambiguous to some extent, especially for those who are not familiar with the field. To address this concern, we revised the manuscript to clarify the scope and limitations of our work.

To provide more accurate descriptions in the revised manuscript, we have clarified that our work contributes to the development of spintronic components within the broader context of neuromorphic computing hardware. We also emphasized that the spintronic synapses and neurons demonstrated in our work are part of a larger system that includes CMOS electronics for tasks such as interface design, data trimming, and data conversion.

Besides, the magnetic field H_z was applied by an external electromagnet, which is used for single device characterization only. For practical application especially integration into chip, all electrical manipulation of device is highly required. Please refer to the response to the Reviewer’s No. 6 Comment for detailed discussions.

In summary, we acknowledge the importance of setting the context properly and clarifying the limitations of our work. We've made substantial revisions to the manuscript to provide a more accurate descriptions of our contributions within the broader framework of neuromorphic computing hardware. We appreciate your valuable feedback, which will help improve the clarity and scientific rigor of the revised manuscript.

Changes highlighted in the revised manuscript:

To follow the referee's advice, at the end of the Introduction and Discussion part in the revised manuscript, we added "It is essential to emphasize that the present research primarily focuses on advancing spintronic components, specifically spintronic synapses and neurons, within a larger framework that integrates CMOS electronics." and "By offering this comprehensive elucidation, it is expected to provide more nuanced insights into the intricate integration of spintronic and CMOS technologies within the realm of neuromorphic computing.", respectively.

3. The second and third sentences of the second paragraph of Introduction are also problematic (by the way, these sentences missed out flash memory and MRAM, which are commercially much more successful than, for example, PRAM). Different memory types have their own advantages and disadvantages: some are slow, some do not last long, some have larger areas, and some have only 1-bit operation. I do not believe that the spintronic devices the authors put forth here will win in all categories, so the above-mentioned sentences are exaggerations that unconditionally promote their own work. Much more scholarly balance is required.

Our response:

We appreciate the reviewer's feedback regarding the second and third sentences of the second paragraph of the Introduction. We apologize for any exaggerations or unbalanced statements made in the original manuscript. We agree that different memory types have their own advantages and disadvantages, and it is important to present a more balanced and accurate view of the field.

In the revised manuscript, we have made substantial revisions to address this concern. We have included a reference to **Supplementary Table 1** (adopted from Supplementary Ref. ⁶) that highlights the advantages of spintronic devices, including MRAM and magnetic DW devices, in terms of endurance, speed, anti-radiation properties, and compatibility with CMOS processes. We acknowledge that these spintronic devices can excel in these aspects. However, we also acknowledge the limitations of spintronic devices, such as their low on/off ratio and limited number of states, which can be detrimental to large-scale integration and neuron network connections. We emphasize the ongoing efforts to improve spintronic device performance, cell structure adjustment, and circuit design optimization to address these challenges. We believe that spintronic devices still hold great promise for neuromorphic computing, despite these limitations.

To provide a more comprehensive view of the competition and a better scholarly balance, we have revised the main text accordingly and added relevant references. We appreciate the reviewer's input, which has helped us improve the accuracy and balance of the manuscript.

Supplementary Table 1. Comparison of artificial neurons and synapses using emerging devices, adopted from Supplementary Ref. 6.

Devices	Schematic ^{a)}	Mechanism	Materials	Pros	Cons	Research status	Artificial neuron	Artificial synapse	Switching energy ^{b)}	On/off ratio ^{b)}	# of States ^{b)}
PCM		Amorphous-crystalline phase change; Ovonic threshold switching	Ge ₂ Sb ₂ Te ₃ (GST)	CMOS compatible	Asymmetric switching Conductance drift	165k cells network to realize MNIST classification	[75–77]	[78–81]	>100 fJ	<10 ³	≈20
Mott Memristor		Metal-to-insulator phase transition	NbO ₂ VO ₂	CMOS compatible	Low resistance in the OFF state	Single (few) device demo	[82]	[83]	>1 pJ	>500	≈2
RRAM		Oxygen filament growth and rupture (filamentary) Defects/ions migration (interface)	Metal oxides 2D materials Perovskite	CMOS compatible High density Low power Gradual G change	Variation Stochasticity	8k array to realize pattern recognition	[84,85]	[30,45,86–95]	>20 fJ	10–100	64–500
CBRAM		Metal filament growth and rupture	Cu/Ag in oxide or chalcogenide	Scalable Low power	Variation Stochasticity Hard to control the filament (gradual G change)	Single device and < 10 × 10 array lab demo	[41,96,97]	[98–101]	>100 fJ	10–240	≈100
FeFET/FTJ		Polarization switching	Doped HfO ₂ Perovskite	CMOS compatible	Hard to realize gradual G change and multilevel (relying on number of domains).	Single device demo	[102]	[94,103]	>100 fJ	45–300	32–100
MRAM/magnetic domain wall device		Magnetization switching	CoFeB MgO	High endurance Fast speed	Limited number of states Small on/off ratio	Single device demo	[104–107]	[108–111]	>10 fJ	2–3	≈2
ECRAM		Electrochemically driven ion intercalation	Metal oxides PEDOT:PSS Graphene	Good symmetry and linearity Gradual G change Multi-states	Write speed Complex unit cell CMOS compatibility	Single device and 3 × 3 array demo	N/A	[29,112–119]	>10 fJ	2–40	50–1000
FETs		Floating gate (FG)/charge trapping based threshold change	Silicon CNT Polymers 2D materials	Technologically more mature	Asymmetric switching Usually high voltage (FG) Speed/endurance	>100k cells for Flash-based demo	N/A	[120–131]	>1 fJ	10–100	20–100

^{a)}The first six inset figures are adapted with permission.^[13–2] Copyright 2018, Springer Nature; ^{b)}The numbers represent the typical values reported in literature for neuromorphic computing.

Changes highlighted in the revised manuscript:

To follow the referee's advice, The second and third sentences of the second paragraph of Introduction has been revised to reflect the complementary nature of NVMs and CMOS electronics in neuromorphic DNN computing. "There are typical reports on the realization of operations in DNNs using different types of NVMs, such as resistive random-access memory (RRAM)^{19,20}, phase change memory (PCM)²¹, ferroelectric RAM (FeRAM)²², flash memory²³, and magnetic RAM (MRAM)²⁴. While these NVMs show promise for neural network applications, they also come with inherent challenges related to nonlinearity, energy efficiency, area overhead, and reliability³. These challenges make it difficult to customize the NVMs, resulting in a loss of learning accuracy and hindrances in implementing specific either synaptic or non-linear activation functions. These issues pose significant confrontations for the practical implementation of NVMs in neural network applications²⁵. Nevertheless, each type of NVM possesses its own advantages and disadvantages, with some being slower, having limited endurance, larger area footprints, or only supporting 1-bit operations. There is pressing need to explore the uniqueness among different memory features and synergistic integration with CMOS in order to achieve optimal performance in neuromorphic DNN computing."

4. When they use their device as a synapse with a proper arrangement of pinning centers and domain walls, they are memory elements. Now when they use their device as a neuron (an activation function) with another suitable arrangement of pinning centers and domain walls, does it cease to be a memory? In the usual operation, the activation function is not supposed to have a memory behavior. The authors should clearly articulate whether the neuron (activation function) is a memory or not, how it is so, and if it is still a memory, whether it will be problematic or not in the scheme of DNN computing.

Our response:

We appreciate the reviewer's question and the opportunity to clarify the behaviors of our developed spin-neuron device in the context of memory and its implications for DNN computing.

The proposed DW-MTJ spin-neuron devices in our work possess the memory capacity due to the intrinsic non-volatile nature of the magnetic domain wall. However, when used as an activation function generator in a neural network, the device retains its state until it is reset or reprogrammed. This memory behavior can be advantageous in certain applications, allowing the neuron device to store information between computations and ensuring the reliable retrieval of information.

However, we understand the concern regarding the memory behavior of the neuron device and its potential impact on DNN computing. In our work, we have considered the issue of memory retention and its implications. We include a reset operation after each programming cycle to ensure the device starts from the initial state for subsequent computations. The time and energy consumption associated with this reset operation are factored into our device performance evaluation.

While the reset operation does introduce additional power consumption, it is a necessary step to ensure the proper functioning of the neuron device in DNN computing. We have taken this into account when evaluating the energy efficiency and area cost of our proposed device compared to CMOS counterparts. Despite the reset operation, neural networks based on non-volatile neuron devices, including ours, can achieve superior energy efficiency and area cost due to their non-

volatility and scalability, as illustrated in *Supporting Information Fig. S9*, i.e., the benchmark of latest representative sigmoidal activation function generator.

Furthermore, it is important to note that non-volatile neuron devices are not limited to DNN computing but also find applications in other types of neural networks (e.g., SNNs), as summarized in **Supplementary Table 2**. In these networks, the reset operation is necessary for proper functioning and is commonly implemented. The use of non-volatile neuron devices in various neural network architectures allows for the simulation of different features of biological neurons.

Supplementary Table 2. Summary of typical neuron devices with memory features in ANN⁷⁻¹³.

Reference	Device	Activation function	Volatile/ Non-volatile	Reset operation	Network structure	Dataset	Experiment/ simulation accuracy
Ref. 7	Magnetic DW Hall-bar	Sigmoid	Non-volatile	need	Two-synapse-layer perceptron networks	MNIST CIFAR-10	Sim. >93%
Ref. 8	Magnetic DW hall-bar	Sigmoid	Non-volatile	need	Tri-layer feed forward ANN	MNIST	Exp.
Ref. 9	Magnetic DW MTJ	Sigmoid	Non-volatile	need	—	—	—
Ref. 10	2 Tansistor-1 Ferroelectric capacitors	ReLU	Non-volatile	need	DNN	Image-net.org	Exp. 93.5%
Ref. 11	2D MoS ₂ Opto-Resistive RAM	Sigmoid, Softplus, ReLU	Non-volatile	need	5 layers fully connected network	MNIST	Sim. 91.6%
Ref. 12	ReMem	ReLU	Non-volatile	need	MLP	MNIST	Sim. 98%
Ref. 13	PCM	Integrate-fire	Non-volatile	need	—	—	—

In summary, our developed DW-MTJ spin-neuron devices possess the memory capacity due to the intrinsic non-volatile nature of the magnetic domain wall. However, the memory behavior does not hinder its application in DNN computing when the device is properly reset after each programming cycle. We have considered the impact of the reset operation on energy consumption and have evaluated the device's performance accordingly. Non-volatile neuron devices, including ours, offer advantages in terms of energy efficiency and area cost compared to CMOS counterparts, making them promising candidates for neuromorphic computing applications.

5. Also can the authors more clearly and easily explain how the nonlinearity arises to make the device perform as what generates the activation function?

Our response:

Thank you for your question regarding the nonlinearity of our proposed device and how it generates the activation function. To follow reviewer's advice, we provide more detailed explanations and references to support our explanation as below.

The nonlinearity in our device arises from the interaction between the magnetic domain wall and the pinning centers within the device structure. When a current is applied to the device, it induces motion of the domain wall. The position of the domain wall within the device determines the resistance and conductance of the device, which in turn influences the output behavior.

The junction resistance of single DW based MTJ is a function of the position of the DW:

$$R_{MTJ} = R_P \left(\frac{x_0}{L} \right) + R_{AP} \left(1 - \frac{x_0}{L} \right) \quad (1)$$

where x_0 , L , R_P (R_{AP}) corresponds to the domain wall position, length of MTJ, resistance of the MTJ when the magnetization of the free and reference layers is entirely in parallel (antiparallel) orientation, respectively. Therefore, the resistance variation when domain wall move to next position x_1 is deduced,

$$\Delta R = R_{MTJ}(x_1) - R_{MTJ}(x_0) = R_{AP} \frac{x_1 - x_0}{L} - R_P \frac{x_1 - x_0}{L} = (R_{AP} - R_P) \frac{\Delta x}{L} \quad (2)$$

A general expression of the sigmoid function is as follows,

$$f(x) = \frac{1}{1 + e^{-x}} \quad (3)$$

When $x > 8$, the sigmoid function converges to 1, and converges to 0 relatively when $x < -8$. Therefore, the approximation of sigmoid function is executed in the range of $x \in (-8, 8)$ ¹⁴. As shown in **Supplementary Fig. 1**, the range $(-8, 8)$ is partitioned into $2N-1$ segments. The N^{th} segment account for Δy_N percent of the whole function range $(0, 1)$.

Supplementary Fig. 1. Segmentation scheme for the sigmoid function.

To realize a sigmoid activation function generator with $2N$ resistance states, $2N$ PCs are formed to divide the domain wall racetrack into $2N-1$ segment with a PC center to center distance of $\Delta y_N \cdot L$ of the N^{th} segment. According to Equation 1, the magnetic resistance variation is a linear function of DW motion distance, therefore, the resistance change for state switching in the proposed sigmoid neuron follow a sigmoid profile.

The specific mechanism behind the nonlinearity can be further elucidated through micromagnetic

simulations using the Mumax³ simulation tool, as shown in **Supplementary Fig. 2**. As demonstrated in **Supplementary Fig. 2a**, the PCs is non-uniformly distributed following the aforementioned design rule. The DW energy distribution versus DW position based on q - φ model in **Supplementary Fig. 2b** explains the origin of nonlinearity in the sigmoidal activation function generator. The designed PCs, marked by pink regions in **Supplementary Fig. 2b** with greater DMI correspond to an energy potential well with the barrier height of $\pi|\Delta D|$. In our design, all PCs shares the same DMI constant, while the DMI constant of the nanowire segments separated by the PCs is related to the segment length. The designed DMI constant distribution can be further explained by the IBE-induced stress effects, as reported by several research work.

Supplementary Fig. 2. Prototype sigmoidal activation function generator. **a** Schematic diagram of proposed spintronic neuron based on DW-pMTJ. **b** DW energy distribution as a function of DW position with different DMI according to q - φ model. Regions in pink rectangles denote PCs with larger DMI constant. Micromagnetic simulation results: **c** DW energy distribution as a function of DW position, and **d** DW position under a sequential SOT current pulse. Regions in blue rectangles refer to PCs.

The DW position dependent DW energy was further analyzed by micromagnetic simulation as shown in **Supplementary Fig. 2c**, which is in line with **Supplementary Fig. 2b**. A non-linear distributed barrier height is observed as indicated by the black dash line, accounting for the non-linear DW motion distance in same duration. The results in **Supplementary Fig. 2d** are well fitted with a shifted sigmoid function as illustrated by the black dashed lines. These simulations can provide insights into the intricate dynamics of the magnetic domain wall and its interaction with the pinning centers, leading to the observed nonlinearity and the generation of the activation function.

6. If the authors created H_z with a current, they do need to specify the amount of current that is needed (I already mentioned this earlier). I had difficulties in finding this information, although the authors might have put it somewhere.

Our response:

Thank you for your question regarding the current required to create the magnetic field component H_z in our device. We apologize for the confusion caused by the lack of specific information on the current values in the manuscript. We appreciate your feedback and have provided the necessary details in the revised manuscript.

In our work, the magnetic field H_z is created using a delicately fabricated electromagnet attached to the MOKE system with electrical transport measurement platform, which allows us to control the nucleation and motion of the domain wall (DW). The current for generating the H_z ranges from tens to few hundreds of milliamperes (mA). And the amplitude of H_z is sensed by an integrated Hall sensor. The specific current values used in the electromagnet were not explicitly mentioned in the manuscript. It is important to note that the purpose of H_z implantation was to demonstrate the functionality of the device. Nevertheless, the exact value of current applied for H_z generation with the precise current values have been provided in the experimental method section in the revised manuscript.

Regarding the nucleation of the domain wall, we can also apply an Oersted field using the metal line above nucleation pad. The current for this process ranges in the 80~90 mA for our micrometer-scale device, which is in line with the typical value reported in the literatures for various device geometries and configurations^{15,16}.

Additionally, we also demonstrated the dynamic control of the DW using purely electrical means, i.e., all-electrical implementation (as elaborated in **Fig. S7**). The supporting videos **S1-S7** accompanying our work provide a detailed demonstration of the electrical control of DW dynamics. In the videos **S4-S7**, the DW motion can be observed to vary at different positions along the channel by varying the number of pulses. This showcases the potential for purely electrical control of DW dynamics.

While our work primarily focuses on the use of an external magnetic field, we acknowledge that further optimization is required for efficient electrical current-driven DW motion. So far, simultaneous realization of electrical manipulation and electrical detection of multi-level resistance state is challenged by the limitation of micro-scale determined by UV lithography, though we had successfully reported the all-electrical or magnetic-field free DW motion and magnetization switching mechanism systematically in our previous studies, e.g., *Adv. Intell. Syst.* 4, 2200028 (2022); *Adv. Sci.* 9, 2203006 (2022); *Nat Commun* 12, 3113 (2021); *US Patent* 18,042,249 (2023); *Materials Today Electronics* 6, 100065 (2023). We have conducted a series of extensive experiments involving materials engineering and optimization of the fabrication process. These investigations were carried out with careful planning, including planned tapeout activities, to ensure the systematic evaluation and refinement of our approach.

Supplementary Fig. 3 a-c dilemma of simultaneous electrical write and detection in the present DW-MTJ based synapse and neuron. d-e Solution to above problem by device size shrinking and increasement of RA to increase junction resistance.

The large device size leads to minimum junction resistance. For example, a typical neuron device is $2\ \mu\text{m}$ in width and $75\ \mu\text{m}$ in length in design, leading to a junction resistance approximately $2\ \Omega$ with Resistance-Area-Product (RA) of $205.879\ \Omega\mu\text{m}^2$ tested through Current-In-Plane Tunneling (CIPT) method, which is in the same magnitude or even smaller than longitudinal resistance of SOT channel. Therefore, as shown in **Supplementary Fig. 3a**, the main part of read current may flow down vertically from top electrode to bottom electrode without flow through the whole domain wall racetrack. To read out multi-states with such a minor junction resistance, a central top electrode covering the domain racetrack was introduced to read junction resistance in the typical pillar MTJ manner, as shown in **Supplementary Fig. 3b**. However, a heavy shunting effect is inevitable and reduce effective current density for domain wall motion remarkably as a result of the central top electrode (Ti20 nm/Au80 nm), as shown in **Supplementary Fig. 3c**. Scaling down device size to nanometer scale or increase barrier layer thickness maybe an effective solution to this, as shown in **Supplementary Fig. 3d**. When scaling down to $50\ \text{nm}$ in width and $1\ \mu\text{m}$ in length with the same RA value, the junction resistance of the nano-scale device is about $4\ \text{k}\Omega$, overwhelming longitudinal resistance of SOT channel. The read current mainly flows in an equivalent resistance network as shown in **Supplementary Fig. 3e**, reading out the multi-level resistance state. While the write current mainly flows between left and right top electrode to maximum effective write current.

Through aforementioned optimization methods from film deposition to device structure and size, the energy-efficient, all electrically manipulated and detected domain wall MTJ device is being realized in a separate future works. Though with some limitations as discussed, we believe that the preliminary results is valid to sustain the main concepts of this work: feasibility of prototype device, potential for all-spin neural network with corroborated compatibility to the CMOS part.

Except that continuing optimization of our EBL process and nano-scale device performance, we have also carried out device fabrication of synapses and neurons crossbar arrays in a more applicable

manner for integration with CMOS peripheral circuits, which is a preliminary exploration of future tape-out plan in standard foundry. The layout and the optical microscopy images of preliminary crossbar arrays is demonstrated in **Supplementary Fig. 4**. We have outlined the specific chip fabrication process to ensure CMOS process-compatible wafer-level system integration and architectural functionality demonstration.

Supplementary Fig. 4. The layout and optical images of the proposed all-spin DW-MTJs synapses and neurons cross bar array for planned tape out processing.

The preliminary experimental demonstration, device optimization analysis, and the extended crossbar array-level devices enrich our presentation and highlight the significance of this work with applicable methods. We believe these detailed illustrations will benefit future studies.

Changes highlighted in the revised manuscript:

We appreciate your feedback. To follow the referee’s advice, in the experiment method part in the revised manuscript, we added “**H_z applied by electromagnet utilizing a current ranging from tens to a few hundreds of milliamperes**” in figure caption.

In the revised *Supporting Information*, we added **Fig. S10 (Supplementary Fig. 3)** and aforementioned limitation discussion “**As shown in Figs. S11 a-c, the simultaneous electrical write and detection is forbad in present micro-scale DW-MTJ based multistate device, owing to non-trivial shunting effect of write current resulted by the central top electrode (Ti/Au 20nm /80 nm) which is crucial for multi-state read for large device size with small junction resistance (~few Ω). However, when the device scaling down to nano-scale or increase the RA as shown in Fig. S11d, the magnetoresistance outweighs longitudinal resistance of write channel in device which eliminate the demand of central top electrode for read operation. As shown in Fig. S11e, the read current injected from left top electrode flows along write channel and then flows down to bottom electrode at each section¹⁷. While the write current mainly flows between left and right top electrode to maximum**

effective write current. Besides, device scaling is also beneficial to energy consumption owing to reduced channel resistance and switching current.” Please refer to the revised *Supporting Information* for more details.

7. The paper feels a bit too long, and I think many technical details can go to supplementary information. Also the abstract is way too detailed to be broadly accessed. The authors should try to write this in a more accessible manner.

Our response:

We appreciate the reviewer's constructive comments and advice. We have carefully considered your suggestions and have conducted substantial revisions and improvements to address the concerns raised.

Firstly, we understand your point about the length of the paper. We recognize that excessive technical details can make it challenging for readers to navigate and comprehend the content effectively. In response to this, we have revised the manuscript and identified sections where certain technical details can be moved to the supplementary information, e.g., the contents of “Simulation parameters, experimental setup, characterization details and Methods, etc.” in the original manuscript were moved into the revised *Supporting Information*. This helps streamline the main text and present the essential findings and concepts in a more concise manner. The supplementary information provides readers with additional technical details for those who desire a deeper understanding of the research.

Additionally, we acknowledge your comments about the level of detail in the **Abstract**. The **Abstract** serves as a concise summary of the paper's key points and is often the first section that readers encounter. To make our research more accessible, we have revised the **Abstract** to ensure that it provides a clear and concise overview of the research. The revised **Abstract** highlights the main objectives, outcomes, and significance of the work, while avoiding excessive technical details.

We appreciate the reviewer's valuable input, as it has helped us improve the clarity and accessibility of our paper. The revisions we have made aim to make our research more widely understandable and impactful. Thank you for bringing these points to our attention, and we look forward to resubmitting the revised manuscript, which reflects the improvements based on your feedback.

Changes highlighted in the revised manuscript:

We appreciate your feedback. To follow the referee’s advice, the entire **Abstract** in the revised manuscript was updated, as “**We report a breakthrough in the hardware implementation of energy-efficient all-spin synapse and neuron devices for highly scalable integrated neuromorphic circuits. Our work demonstrates the successful execution of all-spin synapse and activation function generator using domain wall magnetic tunnel junctions (DW-MTJs). By harnessing the synergistic effects of spin-orbit torque and interfacial Dzyaloshinskii-Moriya interaction (*iDMI*) in selectively etched spin-orbit coupling layers, we achieve a programmable multi-state synaptic device with high reliability. Our first-principles calculations confirm that the reduced atomic distance between $5d$ and $3d$ atoms enhances *iDMI*, leading to stable DW pinning. Our experimental results, supported by visualizing energy landscapes and theoretical simulations, validate the proposed mechanism. Furthermore, we demonstrate a spin-neuron with a sigmoidal activation function, enabling high operation frequency**

up to 20 MHz and low energy consumption of 508 fJ/operation. A neuron circuit design with a compact sigmoidal cell area and low power consumption is also presented, along with corroborated experimental implementation. Our findings highlight the great potential of DW-MTJs in the development of all-spin neuromorphic computing hardware, offering new possibilities for energy-efficient and scalable neural network architectures.”

8. I believe Figures 4 and 5 are experimental highlights, and but then the paper has Figure 7, which is totally based on simulation, but with a very misleading caption of “Circuit-level implementation”. While whether the authors leave Figure 7 in the paper or not is really up to their choice, I personally believe that this has an anti-climactic effect that draws the reader’s attention from what they should focus on (the two key experimental demonstrations for synapses and neurons). But at the least, the authors must be very explicit that Figure 7 is purely based on simulation; while they do state that, it is not immediately obvious with the presently ambiguous descriptions. Certainly, wordings like “Circuit-level implementation” must not be used: they are certainly too misleading.

Our response:

Thank you for your valuable feedback regarding **Figs. 4, 5, and 7** in original manuscript. We appreciate the reviewer's constructive comments and have carefully considered their suggestions for improvement.

We acknowledge the concern raised about the potential anti-climactic effect of **Fig. 7**, which is based on simulation rather than experimental data in the original manuscript. We understand that the caption "*Circuit-level implementation*" may be misleading and draw attention away from the key experimental demonstrations presented in **Figs. 4 and 5**, which are the primary focus of our paper.

In response to this feedback, we have made substantial revisions to the paper. While we have chosen to retain **Fig. 7** because that it is the circuit-level corroboration of findings from **Figs. 4 and 5** to verify the compatibility of the spin devices with CMOS circuits. Besides, we have updated the data to include additional experimental results and demonstrations in the **updated Fig. 7**. This update provides a more comprehensive understanding of the topic and addresses the interest expressed by another reviewer regarding the simulation and subcircuit demonstration aspects of our work. Furthermore, we have taken your advice and revised the caption of **updated Fig. 7** to ensure clarity and explicitness.

Additionally, we have worked on enhancing the readability and logical flow of the paper. We have restructured the narrative to provide a more compact and coherent scientific storyline, emphasizing the key experimental demonstrations for spintronic synapses and neurons in **Figs. 4, 5 and 7**. Our aim is to guide the readers’ focus towards these crucial experimental findings and their significance in the context of our research.

We appreciate the reviewer's valuable inputs, which have helped us improve the accuracy and clarity of our paper. The revisions we have made, including the updates to **Fig. 7**, the captions, and the improvements in the paper's structure and presentation discussions, aim to address the concerns raised and enhance the overall quality of the manuscript. Thank you for bringing these points to our attention, and we look forward to your further support and comments on the revised manuscript, reflecting the improvements based on your comments.

Updated Fig. 7 Circuit simulation and experimental verification of spin neuron circuit. **a** Schematic of a simplified ANN network comprising 16×16 DW-MTJ synapses and 16×1 16-state DW-MTJ neurons. **b**. Transient simulation results of the spin neuron circuit. **c** Hardware implementation of with developed 4 binary SOT-MTJs as synapse and 1 spin-neuron device (right panel). **d** Waveform of the pre-neuron signal output by MCU and corresponding output of the operational amplifier. **e** MOKE Kerr images captured after each pre-neuron signal applied.

Changes highlighted in the revised manuscript:

With a Supporting Information Video S7 to visualize aforementioned experiment implementation, we added an extensive discussions in the section of **Simulation and experimental verification of spin neuron circuit** in the revised manuscript: “A simulation of a hybrid MTJ/CMOS system that incorporated 16×16 DW-MTJ synapses and a 16×1 16-state DW-MTJ neurons was performed (parameters listed in Table S2). Furthermore, to ensure the reliability of developed devices, we conducted a validation of the system's performance using a neuromorphic hardware implementation of representative arrays. The results of both the simulation and experiments are presented in **Fig. 7**. The purpose of such complementary study is to assess the feasibility of employing all-spin synapses and a sigmoidal activation function generator, as well as to evaluate the compatibility of our developed devices with CMOS technology.”.

Supplementary References

- 1 Sengupta, A., Shim, Y. & Roy, K. Proposal for an All-Spin Artificial Neural Network: Emulating Neural and Synaptic Functionalities Through Domain Wall Motion in Ferromagnets. *IEEE Trans. Biomed. Circuits. Syst.* **10**, 1152-1160 (2016).
- 2 Sengupta, A. & Roy, K. A Vision for All-Spin Neural Networks: A Device to System Perspective. *IEEE Trans. Circuits Syst. I* **63**, 2267-2277 (2016).
- 3 Zhang, D. *et al.* All Spin Artificial Neural Networks Based on Compound Spintronic Synapse and Neuron. *IEEE Trans. Biomed. Circuits. Syst.* **10**, 828-836 (2016).
- 4 Wu, M. H. *et al.* Extremely Compact Integrate-and-Fire STT-MRAM Neuron: A Pathway toward All-Spin Artificial Deep Neural Network. In *2019 Symposium on VLSI Technology*. T34-T35.
- 5 Wu, M. H. *et al.* Compact Probabilistic Poisson Neuron Based on Back-Hopping Oscillation in STT-MRAM for All-Spin Deep Spiking Neural Network. In *2020 IEEE Symposium on VLSI Technology*. 1-2.
- 6 Tang, J. *et al.* Bridging Biological and Artificial Neural Networks with Emerging Neuromorphic Devices: Fundamentals, Progress, and Challenges. *Adv. Mater.* **31**, 1902761 (2019).
- 7 Yang, S. *et al.* Integrated neuromorphic computing networks by artificial spin synapses and spin neurons. *NPG Asia Mater.* **13**, 4057 (2021).
- 8 Zhou, J. *et al.* Spin-Orbit Torque-Induced Domain Nucleation for Neuromorphic Computing. *Adv. Mater.* **33**, e2103672 (2021).
- 9 Siddiqui, S. A. *et al.* Magnetic Domain Wall Based Synaptic and Activation Function Generator for Neuromorphic Accelerators. *Nano Lett.* **20**, 1033-1040 (2020).
- 10 Hsieh, E. R. *et al.* The First Embedded 14nm FeFinFET NVM: 2T1CFE Array as Electrical Synapses and Activations for High-performance and Low-power Inference Accelerators. In *2021 Symposium on VLSI Technology*. 1-2.
- 11 Xu, Z. *et al.* Reconfigurable nonlinear photonic activation function for photonic neural network based on non-volatile opto-resistive RAM switch. *Light: Sci. Appl.* **11**, 288 (2022).
- 12 Wang, Z. *et al.* Self-Activation Neural Network Based on Self-Selective Memory Device With Rectified Multilevel States. *IEEE Trans. Electron Devices* **67**, 4166-4171 (2020).
- 13 Oh, S. *et al.* Energy-efficient Mott activation neuron for full-hardware implementation of neural networks. *Nat. Nanotechnol.* **16**, 680-687 (2021).
- 14 Qin, Z. *et al.* A Novel Approximation Methodology and Its Efficient VLSI Implementation for the Sigmoid Function. *IEEE Transactions on Circuits and Systems II: Express Briefs* **67**, 3422-3426 (2020).
- 15 Alejos, O., Raposo, V., Sanchez-Tejerina, L. & Martinez, E. Efficient and controlled domain wall nucleation for magnetic shift registers. *Sci. Rep.* **7**, 11909 (2017).
- 16 Phung, T. *et al.* Highly efficient in-line magnetic domain wall injector. *Nano Lett.* **15**, 835-841 (2015).
- 17 Kumar, D. *et al.* Ultralow Energy Domain Wall Device for Spin-Based Neuromorphic Computing. *ACS Nano* **17**, 6261-6274 (2023).

REVIEWERS' COMMENTS

Reviewer #1 (Remarks to the Author):

The authors have carefully addressed all my concerns. The paper is in a much better shape now.

Reviewer #2 (Remarks to the Author):

The authors have done a very rigorous job in good spirit to address all of my previous comments, and the presentation of the paper has now a much better scholarly balance. I suggest the publication of the paper.

Response to the referees

Manuscript ID: NCOMMS-23-45616A

TITLE: Domain wall magnetic tunnel junction-based artificial synapses and neurons for all-spin neuromorphic hardware

REVIEWERS' COMMENTS

Reviewer #1 (Remarks to the Author):

The authors have carefully addressed all my concerns. The paper is in a much better shape now.

Our response:

We sincerely appreciate the positive evaluation provided by reviewer regarding our revised manuscript. It is gratifying to hear that the reviewer recognizes the improvements made and acknowledges the significant enhancement in the quality of the paper. We are truly grateful for the opportunity to refine our work and maintain a commitment to excellence. We extend our sincere thanks to the reviewer for their valuable input, which has contributed to the overall enhancement of our manuscript for publication.

Reviewer #2 (Remarks to the Author):

The authors have done a very rigorous job in good spirit to address all of my previous comments, and the presentation of the paper has now a much better scholarly balance. I suggest the publication of the paper.

Our response:

We sincerely appreciate the positive feedback and endorsement of our revised manuscript provided by the reviewer. We are delighted that our efforts to address the previous comments have resulted in a significantly improved scholarly balance and presentation of the paper. We extend our sincere thanks to the reviewer for their valuable input and support throughout the review process with recommendation for publication.